# Lrp10 suppresses IL7R limiting CD8 T cell homeostatic expansion and anti-tumor immunity

Jamie Russell[1,4], Luming Chen [ID][2,4], Aijie Liu[1], Jianhui Wang[1], Subarna Ghosh [ID][1], Xue Zhong [ID][1], Hexin Shi[1], Bruce Beutler [ID][1] & Evan Nair-Gill [ID][1,3 ✉]

## Abstract

**Signals emanating from the T-cell receptor (TCR), co-stimulatory receptors, and cytokine receptors each influence CD8 T-cell fate. Understanding how these signals respond to homeostatic and microenvironmental cues can reveal new ways to therapeutically direct T-cell function. Through forward genetic screening in mice, we discover that loss-of-function mutations in *LDL receptor-related protein 10* (*Lrp10*) cause naive and central memory CD8 T cells to accumulate in peripheral lymphoid organs. *Lrp10* encodes a conserved cell surface protein of unknown immunological function. T-cell activation induces Lrp10 expression, which post-translationally suppresses IL7 receptor (IL7R) levels. Accordingly, *Lrp10* deletion enhances T-cell homeostatic expansion through IL7R signaling. *Lrp10*-deficient mice are also intrinsically resistant to syngeneic tumors. This phenotype depends on dense tumor infiltration of CD8 T cells, which display increased memory cell characteristics, reduced terminal exhaustion, and augmented responses to immune checkpoint inhibition. Here, we present Lrp10 as a new negative regulator of CD8 T-cell homeostasis and a host factor that controls tumor resistance with implications for immunotherapy.**

**Keywords** CD8 T Cell; Homeostatic Expansion; Central Memory; Anti-tumor Immunity; IL7R
**Subject Categories** Cancer; Immunology; Signal Transduction

## Introduction

CD8 T cells circulate through peripheral lymphoid organs until encountering their cognate antigen. Antigen-induced activation leads to clonal expansion, expression of effector molecules, and migration to inflamed tissues. Clearance of the antigenic stimulus is followed by apoptosis of most antigen-specific cells (D'Cruz et al, 2009). A small number of memory cells remain that are long-lived, variably capable of self-renewal, and can rapidly respond to subsequent challenges (Jameson and Masopust, 2018). When antigenic stimulation persists, for example within the tumor microenvironment (TME), or during a chronic viral infection, the transition from the effector phase to the memory phase is corrupted. In these instances, CD8 T-cell function is impaired through a differentiation process known as "exhaustion" wherein subpopulations of tumor-reactive clones with memory and stem-like features continuously propagate a terminally exhausted pool (Zehn et al, 2022). An array of transcription and epigenetic factors promote or oppose T-cell exhaustion (Belk et al, 2022; Kaech and Cui, 2012). In addition, signals transmitted through the T-cell receptor (TCR), inhibitory and activating co-receptors, and cytokine receptors influence CD8 T-cell fate according to extrinsic cues (Giles et al, 2023; Huang and August, 2015; Wei et al, 2019). How these externally derived signals integrate with cell-intrinsic gene regulatory programming remains uncertain. Therefore, identifying mechanisms that control the intensity and duration of externally derived signals could inform CD8 T cell-based immunotherapies.

To define new mechanisms that control T-cell homeostasis and differentiation, we performed a forward genetic screen in randomly mutagenized mice that measured the proportions of T cells circulating in the peripheral blood. Here, we report that a gene called *LDL receptor-Related Protein 10* (*Lrp10*) plays an important role in the homeostasis and differentiation of peripheral CD8 T cells through its effects on the interleukin 7 receptor (IL7R). *Lrp10* encodes a putative endocytic receptor that is a member of the LDL Receptor-related Protein (LRP) superfamily (Sugiyama et al, 2000). Most LRPs bind and internalize diverse ligands through large extracellular domains that contain cysteine-rich LDL ligand-binding domains (LBD) and Epidermal Growth Factor (EGF) homology domains (Lane-Donovan et al, 2014). In contrast, Lrp10, along with Lrp3 and Lrp12, form a distinct subfamily of orphan LRPs with relatively small extracellular regions that contain LBDs and C1r/C1s, Uegf, Bmp1 (CUB) domains. Lrp10 was shown previously to facilitate intracellular vesicle trafficking in neuronal cells and astrocytes in cell culture (Brodeur et al, 2012). It also negatively regulated the growth of myeloid leukemia cells in mice (Ramakrishnan et al, 2020). Currently, there is little knowledge of its function in vivo and it has no previously described role in

[1]Center for the Genetics of Host Defense, University of Texas Southwestern Medical Center, 5323 Harry Hines Blvd, Dallas, TX 75390-8505, USA. [2]Medical Scientist Training Program, University of Texas Southwestern Medical Center, 5323 Harry Hines Blvd, Dallas, TX 75390-8505, USA. [3]Department of Internal Medicine, Division of Rheumatic Diseases, University of Texas Southwestern Medical Center, 5323 Harry Hines Blvd, Dallas, TX 75390-8505, USA. [4]These authors contributed equally: Jamie Russell, Luming Chen. ✉E-mail: Evan.Nair-Gill@UTSouthwestern.edu

immune homeostasis. By deleting *Lrp10* in mice, we have discovered that Lrp10 prevents accumulation of naive and memory CD8 T cells in secondary lymphoid organs, limits IL7R expression, suppresses T-cell homeostatic expansion, and impairs anti-tumor immune responses.

## Results

### Forward genetic screening reveals *Lrp10* is critical for normal CD8 T-cell and NK cell homeostasis in mice

To identify new determinants of immune homeostasis, we screened the peripheral blood of mice mutagenized with N-ethyl-N-nitrosourea (ENU) with flow cytometry (Wang et al, 2015; Xu et al, 2021). Several mice from a single pedigree showed an increased proportion of CD8 T cells and a decreased proportion of NK1.1+ cells, a phenotype that we named *chowmein*. Automated meiotic mapping linked the *chowmein* phenotype to a missense mutation in *LDL receptor-related protein 10* (*Lrp10*) using a recessive model of inheritance (Fig. 1A,B).

Lrp10 possesses an extracellular domain (ECD) containing two CUB ligand-binding domains interspersed with LBDs, a single-pass transmembrane domain (TM), and a proline-rich intracellular domain (ICD) (Fig. 1C). The *chowmein* allele encoded an aspartate to tyrosine substitution at position 246 (D246Y) in the second CUB domain of the ECD. Lrp10 was expressed at low levels in unstimulated CD8 T cells and its expression increased with T-cell activation (Fig. 1D).

To verify that the observed phenotypes were caused by the loss of Lrp10, we used CRISPR-Cas9 to create a constitutive knockout allele of *Lrp10*. *Lrp10* knockout mice (*Lrp10*$^{-/-}$) were born at the expected Mendelian ratio, appeared outwardly normal, and were viable and fertile. *Lrp10*$^{-/-}$ mice showed an increased proportion of CD8 T cells and a reduction in NK1.1+ cells in the peripheral blood, confirming that loss of Lrp10 function was responsible for the *chowmein* phenotype (Fig. 1E). Lethally irradiated *Rag2*$^{-/-}$ mice reconstituted with *Lrp10*$^{-/-}$ bone marrow had increased numbers of peripheral CD8 T cells and reduced numbers of NK1.1+ cells compared to those receiving *Lrp10*$^{+/+}$ bone marrow, indicating that these phenotypes were hematopoietic-intrinsic (Fig. 1F).

### *Lrp10*$^{-/-}$ mice accumulate naive and central memory CD8 T cells in a TCR repertoire-dependent manner

Spleens from *Lrp10*$^{-/-}$ mice harbored increased absolute numbers of CD8 T cells reflecting what we observed in the peripheral blood (Fig. 1G). Among the other major adaptive immune populations, there was no difference in the numbers of CD4 + T cells or B220 + B cells. Amongst the innate immune populations, there were decreased numbers of natural killer cells (NK, Lin-CD11b + /−CD127-NK1.1 + NKp46 + ) but similar numbers of splenic innate lymphoid cells (ILCs, Lin-CD11b-CD127 + NK1.1 + NKp46 + ) (Spits et al, 2016). Within the splenic myeloid populations, there were no differences in the numbers of monocytes/macrophages, neutrophils, or dendritic cells between *Lrp10*$^{+/+}$ and *Lrp10*$^{-/-}$ mice.

We first considered that the increase in peripheral CD8 T cells in *Lrp10*$^{-/-}$ mice might be due to preferential skewing toward the CD8+ lineage during thymic selection. However, thymic cellularity was the same between *Lrp10*$^{+/+}$ and *Lrp10*$^{-/-}$ mice with respect to double negative, double positive, and CD4 and CD8 single-positive thymocytes (Fig. EV1A).

The peripheral CD8 T-cell population in mice contains three broad subpopulations defined by expression of the lymph node homing receptor CD62L and the tissue homing receptor CD44: naive cells that have not been exposed to antigen (T$_N$, CD62L + CD44-), central memory cells (T$_{CM}$, CD62L + CD44 + ) that circulate through secondary lymphoid organs, and effector memory cells that patrol peripheral tissues and exhibit lower levels of lymphoid recirculation (T$_{EM}$, CD62L-CD44 + ) (Nolz et al, 2011). *Lrp10*$^{-/-}$ mice showed a slight increase in the number of CD8 T$_N$ cells compared to *Lrp10*$^{+/+}$ mice (Fig. 1H–L). Interestingly, *Lrp10*$^{-/-}$ mice exhibited a two- to threefold expansion in the CD8 T$_{CM}$ population. In contrast, the number of CD8 T$_{EM}$ cells was the same between *Lrp10*$^{+/+}$ and *Lrp10*$^{-/-}$ spleens. We did not observe any differences in the distribution of the memory subpopulations in *Lrp10*$^{-/-}$ CD4 T cells (Fig. EV1B).

CD8 T$_{CM}$ cells appearing in unimmunized mice can arise from endogenous/self-antigen exposure and homeostatic expansion driven by cytokines (Fry and Mackall, 2005; Jameson and Masopust, 2018; White et al, 2017). To define the role of TCR specificity in the accumulation of *Lrp10*$^{-/-}$ CD8 T$_{CM}$ cells, we crossed *Lrp10*$^{-/-}$ mice to the OT-1 TCR transgenic strain in which the majority of CD8 T cells express a TCR specific for ovalbumin (ova). Compared to *Lrp10*$^{-/-}$ mice with a diverse TCR repertoire, *Lrp10*$^{-/-};OT-1* mice had a higher number of T$_N$ and a lower number of T$_{CM}$ cells (Fig. 1H–L). The ratio of T$_{CM}$ to T$_N$ cells was normalized in *Lrp10*$^{-/-};OT-1* compared to *Lrp10*$^{-/-}$ mice. Restricting the TCR repertoire did not change the number of CD8 T$_{EM}$ cells in the spleen. These results show that restricting the CD8 TCR repertoire imparts a block in the conversion of *Lrp10*$^{-/-}$ T$_N$ cells to T$_{CM}$ cells and suggests that TCR responsiveness to endogenous/self-antigens is important for the accumulation of T$_{CM}$ cells in *Lrp10*$^{-/-}$ mice. In agreement with this idea, we found that *Lrp10*$^{-/-}$ CD8 T$_{CM}$ and T$_{EM}$ cells, but not T$_N$ cells, showed increased levels of CD5 expression, a negative regulator of TCR signaling whose expression correlates with increased reactivity to self-peptide/MHC-I (Fig. EV2A). *Lrp10*$^{-/-}$ CD4 T-cell subsets did not show similar increases in CD5 expression.

### Normal antigen-specific proliferation and CD8 cytotoxic responses in *Lrp10*$^{-/-}$ CD8 T cells

We first hypothesized that *Lrp10* deletion sensitized TCR signaling to antigens that were present in low amounts or that had low TCR affinities. However, *Lrp10*$^{+/+};OT-1* and *Lrp10*$^{-/-};OT-1* cells showed similar proliferative responses to high and low doses of ova in vitro (Fig. EV2B). In addition, adoptively transferred *Lrp10*$^{+/+};OT-1* and *Lrp10*$^{-/-};OT-1* cells showed similar proliferative responses in vivo after immunization with SIINFEKL (a high-affinity peptide antigen for the OT-1 TCR) or SIITFEKL (a peptide antigen with ~100-fold lower TCR affinity) (Fig. EV2C).

We next assessed how the loss of Lrp10 affected cytotoxic responses. *Lrp10*$^{+/+}$ and *Lrp10*$^{-/-}$ mice showed similar levels of killing SIINFEKL-pulsed target cells after immunization with ova and alum adjuvant (Fig. EV2D). Consistent with lower circulating NK cell numbers, *Lrp10*$^{-/-}$ mice had a mild defect in killing MHC-Class I-deficient target cells (Fig. EV2E).

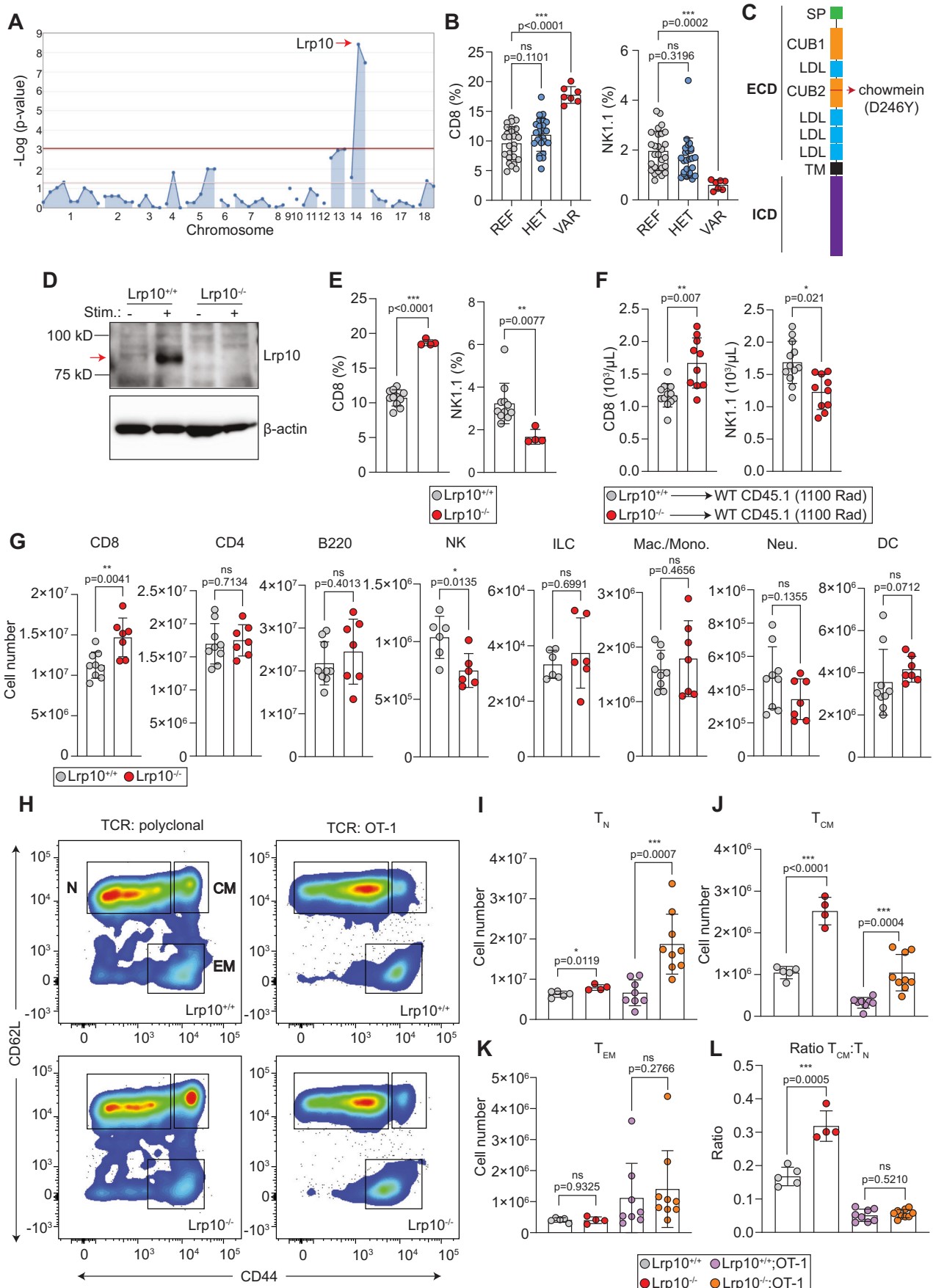

◄ **Figure 1.  Lrp10 deletion increases CD8 T cells.**

(A) Manhattan plot showing linkage between an ENU-induced point mutation in *Lrp10* and increased peripheral CD8 T cells. (B) Frequency of peripheral CD8 T cells and NK cells in the *chowmein* pedigree. (C) Domain structure of Lrp10 and location of the ENU-induced substitution. (D) Expression of Lrp10 in resting and stimulated splenic CD8 T cells. (E) Frequency of CD8 T and NK cells in the peripheral blood of CRISPR *Lrp10$^{-/-}$* mice. (F) Number of peripheral CD8 T cells and NK cells from lethally irradiated mice transplanted with *Lrp10$^{+/+}$* or *Lrp10$^{-/-}$* bone marrow. (G) Numbers of the major splenic immune lineages. NK cells were defined as Lin-CD11b + /−IL7R-NK1.1 + NKp46+ and ILCs were defined as Lin-CD11b-IL7R + NK1.1 + NKp46 + . Lin: Ter119, CD3, B220, CD11c. Within the Ter119-CD3-B220-NK1.1− fraction, monocytes/macrophages were defined as CD11c-CD11b+SSC-low (both Gr1+ and Gr1−), neutrophils were defined as CD11c-CD11b + Gr1+SSC-high, and dendritic cells were defined as CD11c + . (H) Representative FACS plot of CD8 T-cell subpopulations from *Lrp10$^{+/+}$* and *Lrp10$^{-/-}$* mice harboring a polyclonal repertoire and a restricted repertoire (OT-1). (I–K) Quantification of CD8 T-cell subpopulations dependent on TCR repertoire. (L) Ratio of CD8 T$_{CM}$:T$_N$ cells. Data information: In bar graphs, symbols show individual mice (biological replicates), horizontal bars show the mean, and error bars show SD. (E) Results were replicated three times on separate CD8 T-cell samples. (F) Results were replicated in two separate bone marrow transplantation experiments. *P* values were calculated by one-way ANOVA with Dunnett's multiple comparisons test (B), Mann–Whitney test (J–L), or two-tailed unpaired *t* test (E–G, I, J). Significant *P* values were flagged as follows: *$P < 0.05$, **$P < 0.01$, ***$P < 0.001$. *P* values >0.05 were considered to be not significant (ns). Source data are available online for this figure.

We performed CD8 cytotoxicity assays at timepoints after ova-alum immunization and found that *Lrp10* deletion did not impart higher levels of target cell killing over time (Fig. EV2F). Moreover, both *Lrp10$^{+/+}$* and *Lrp10$^{-/-}$* mice responded similarly to a boost with ova performed 90 days after immunization. Overall, these data show that although *Lrp10* deletion promotes the differentiation and accumulation of CD8 T$_{CM}$ cells, it does not enhance TCR sensitivity or promote unrestrained CD8 T-cell cytotoxicity or cytotoxic memory.

### *Lrp10* limits IL7R expression and T-cell homeostatic proliferation

CD8 T$_N$ and T$_{CM}$ require IL7R signaling for differentiation and survival (Carrette and Surh, 2012; Schluns et al, 2000). Based on the accumulation of T$_N$ and T$_{CM}$ in *Lrp10$^{-/-}$* mice, we next analyzed cell surface expression of IL7R (Fig. 2A). CD8 T$_N$ and T$_{CM}$ from *Lrp10$^{-/-}$* mice showed ~30% and ~50% increased cell surface IL7R expression, respectively. CD8 T$_{EM}$ cells displayed a bimodal expression pattern of IL7R. *Lrp10$^{-/-}$* CD8 T$_{EM}$ cells showed ~35% increased cell surface IL7R within the IL7R+ subpopulation. More IL7R was also present on the surface *Lrp10$^{-/-}$* CD4 T-cell subsets, although not to the levels seen on *Lrp10$^{-/-}$* CD8 T cells (Fig. EV3A). ILCs are NK1.1+ cells that express IL7R in peripheral tissues and lymphoid organs (Spits et al, 2016). *Lrp10$^{-/-}$* splenic ILCs also expressed higher levels of cell surface IL7R (Fig. EV3B).

IL7R signaling is critical for thymic T-cell development where it is expressed initially at the DN stage, turned off at the DP stage, and again expressed during the selection of single-positive CD4 and CD8 T cells (Singer et al, 2008). There was no substantial difference in IL7R expression in *Lrp10$^{-/-}$* DN, DP, or CD4 single-positive T cells (Fig. EV3C). *Lrp10$^{-/-}$* single-positive CD8 T cells showed a small increase in IL7R expression compared to *Lrp10$^{+/+}$* cells, although this difference was much less compared to peripheral CD8 T cells.

IL7 binds IL7R to activate STAT5, upregulate Bcl2, and promote T-cell survival (Mazzucchelli and Durum, 2007; Rochman et al, 2009). Splenic *Lrp10$^{-/-}$* CD8 T cells showed increased phosphory-lated STAT5 and Bcl2 expression, consistent with enhanced basal IL7R signaling (Fig. 2B). Repeated injections of IL7/anti-IL7 immune complexes (IL7 IC) into mice has been shown to stimulate IL7R to increase T-cell numbers (Boyman et al, 2008). We found that a single injection of IL7 IC caused preferential expansion of

T$_{CM}$ and T$_{EM}$ in *Lrp10$^{-/-}$* spleens, indicating that *Lrp10* deletion sensitized these populations to exogenous IL7 (Fig. 2C).

IL7R signaling promotes T-cell homeostatic proliferation during lymphopenia (Kimura et al, 2013; Tan et al, 2001). To further examine IL7 responsiveness in *Lrp10$^{-/-}$* CD8 T cells, we tested their ability to proliferate under lymphopenic conditions. Naive splenic CD8 T cells were harvested from *Lrp10$^{+/+}$* and *Lrp10$^{-/-}$* mice, differentially labeled with cell proliferation dye, and transplanted in equal numbers into syngeneic sub-lethally irradiated recipients (Fig. 2D). *Lrp10$^{-/-}$* CD8 T cells showed a rapid increase in cell proliferation starting three days post-transfer. By 7 days, ~90% of cells had undergone at least one round of cell division. In contrast, *Lrp10$^{+/+}$* CD8 T cells showed lower rates of homeostatic expansion at early timepoints and by day 7, ~60% cells had undergone at least one round of division. Cells transferred into lymphocyte-replete recipients showed no dye dilution, indicating that *Lrp10$^{-/-}$* CD8 T cells did not spontaneously proliferate.

CD4 T cells are present in normal numbers in *Lrp10$^{-/-}$* mice and display slightly increased levels of IL7R. Consistent with these findings, *Lrp10$^{-/-}$* CD4 T cells showed mildly enhanced homeo-static proliferation upon transfer into sub-lethally irradiated recipients (Fig. EV3D).

IL7R signaling and TCR signaling arising from self-peptide/MHC interactions each control T-cell homeostatic proliferation (Kawabe et al, 2021). We next dissected how these distinct signals contributed to *Lrp10$^{-/-}$* CD8 T-cell homeostatic expansion. We labeled *Lrp10$^{+/+}$* (CD45.1) and *Lrp10$^{-/-}$* (CD45.2) naive CD8 T cells with cell proliferation dye and adoptively transferred them into sub-lethally irradiated syngeneic mice. Recipient mice were then injected with an IL7R-blocking antibody or an isotype control antibody (Fig. 2E). *Lrp10$^{-/-}$* CD8 T cells from mice injected with the isotype control antibody showed the typical increased expansion phenotype. Conversely, anti-IL7R completely blocked the proliferation of *Lrp10$^{+/+}$* and *Lrp10$^{-/-}$* CD8 T cells. *Lrp10$^{-/-}$* CD8 T cells were recovered at a higher frequency from isotype control-injected mice and this competitive advantage was neutra-lized through IL7R blockade.

To test the effect of TCR signaling on CD8 T-cell homeostatic expansion, we adoptively transferred *Lrp10$^{+/+}$* (CD45.1) and *Lrp10$^{-/-}$* (CD45.2) naive CD8 T cells into sub-lethally irradiated syngeneic mice that lacked MHC-I expression (*Beta-2 microglobu-lin* knockout, *B2m$^{-/-}$*). Transfer of CD8 T cells into *B2m$^{-/-}$* mice substantially limited the homeostatic expansion of both *Lrp10$^{+/+}$*

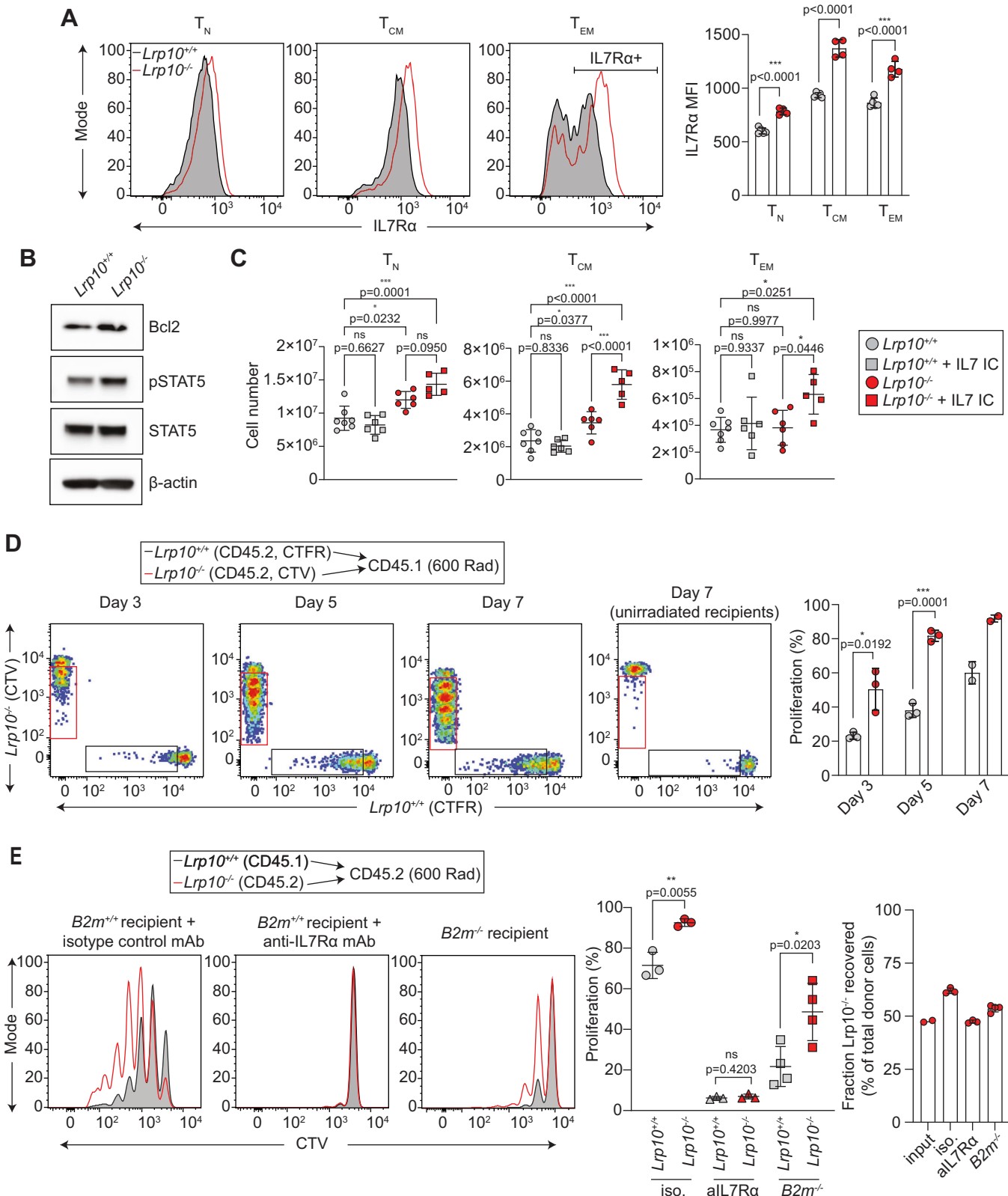

◄ **Figure 2. *Lrp10* deletion increases IL7R expression and function.**

(A) IL7R surface expression on splenic CD8 T-cell subpopulations. (B) Expression of markers for IL7R signaling in resting splenic CD8 T cells. (C) Effect of IL7/anti-IL7 immune complex (IC) administration on splenic CD8 T-cell populations. (D) Homeostatic expansion of differentially labeled naive *Lrp10*$^{+/+}$ (CTFR) and *Lrp10*$^{-/-}$ (CTV) CD8 T cells in lymphopenic hosts. Proliferation was determined from the percentage of cells that underwent at least one cell division. (E) Homeostatic expansion of naive *Lrp10*$^{+/+}$ (CD45.1) and *Lrp10*$^{-/-}$ (CD45.2) CD8 T cells injected into sub-lethally irradiated *B2m*$^{+/+}$ or *B2m*$^{-/-}$ (CD45.2) recipients. Mice were treated with anti-IL7R or isotype control antibodies as indicated. Homeostatic competitiveness was assessed based on the fraction of *Lrp10*$^{+/+}$ (gray bar) and *Lrp10*$^{-/-}$ (red bar) cells recovered on D7 compared to the input. Data information: In bar graphs, symbols show individual mice (biological replicates), horizontal bars show the mean, and error bars show SD. (B) Results were replicated twice on separate CD8 T-cell samples. (C) Results are combined from two experiments with at least two mice per treatment group. (D) Results were replicated in three separate experiments with at least two recipient mice per timepoint. (E) Data were combined from one IL7R blockade experiment and one *B2m*$^{-/-}$ transplantation experiment with the following n: input $n = 2$, isotype $n = 3$, anti-IL7R $n = 3$, *B2m*$^{-/-}$ $n = 4$. P values were calculated by two-tailed unpaired t test (A, D, E) or one-way ANOVA with Tukey's multiple comparisons test (C). Significant P values were flagged as follows: *$P < 0.05$, **$P < 0.01$, ***$P < 0.001$. P values > 0.05 were considered to be not significant (ns). Source data are available online for this figure.

and *Lrp10*$^{-/-}$ CD8 T cells (Fig. 2E). However, *Lrp10*$^{-/-}$ CD8 T cells continued to display higher levels of dye dilution and were recovered at increased frequencies. These findings show that increased homeostatic expansion of *Lrp10*$^{-/-}$ CD8 T cells depends entirely on IL7R signaling. While TCR and IL7R signaling combine to augment the homeostatic expansion of *Lrp10*$^{-/-}$ CD8 T cells, they can proliferate at a reduced capacity in the absence of TCR-MHC-I interactions.

## Lrp10 post-translationally suppresses IL7R expression

We next investigated how Lrp10 modulated IL7R expression. *IL7R* mRNA levels were the same in *Lrp10*$^{+/+}$ and *Lrp10*$^{-/-}$ CD8 T cells, indicating that Lrp10 reduced IL7R cell surface expression through a post-transcriptional mechanism (Fig. 3A). We used retroviruses encoding Lrp10, or an empty vector control, to complement *Lrp10*$^{-/-}$ CD8 T cells (Fig. 3B). Re-introducing Lrp10 reduced cell surface IL7R on *Lrp10*$^{-/-}$ CD8 T cells, suggesting a direct link between Lrp10 and IL7R protein expression. Confirming this finding, heterologous expression of Lrp10 and IL7R in HEK 293T cells showed that Lrp10 suppressed IL7R expression in a dose-dependent manner (Fig. 3C).

Lrp10 is a single-pass type-1 transmembrane protein with an ECD composed of LDL ligand-binding and CUB domains, a TM, and an ICD of unknown function. We co-expressed recombinant Lrp10 variants that lacked each of these domains with IL7R (Fig. 3D). Deletion of the ICD and TM, leaving only a secreted ECD, completely rescued the suppressive effect of Lrp10 on IL7R expression. In contrast, co-expression of Lrp10-ICD/TM potently suppressed IL7R expression. Additional deletion of the TM domain, leaving only the ICD, partially reversed the suppressive effect on IL7R expression. Together, these data indicate that the Lrp10-ICD mediates suppression of IL7R and that this effect is potentiated by its transmembrane localization.

We hypothesized that Lrp10 bound to IL7R to limit its expression. The heterologous expression of epitope-tagged Lrp10 and IL7R in HEK 293T cells followed by co-immunoprecipitation (co-IP) showed an interaction between full-length IL7R (IL7R-FL) and Lrp10-FL (Fig. 3E). In addition, co-IP demonstrated an interaction between IL7R-FL and both the Lrp10-ECD and ICD, although binding was qualitatively higher with the Lrp10-ICD constructs. Deletion of the IL7R ECD strongly attenuated binding to each of the Lrp10 domains indicating that Lrp10 associates with IL7R in an IL7R ECD-dependent manner. The ability of both the Lrp10-ECD and ICD to bind to the IL7R ECD does not reconcile

with a model of direct protein-protein interaction and instead suggests that Lrp10 and IL7R may each be associated with a shared protein complex.

We noted that co-expression with Lrp10-FL, Lrp10-ICD/TM, and Lrp10-ICD reduced the levels of IL7R migrating at ~60 kD and instead showed multiple IL7R bands of lower molecular weights (Fig. 3C). While its predicted molecular weight is 52 kD, IL7R is highly glycosylated which positively affects its ability to bind IL7 (McElroy et al, 2009). We speculated that the lower molecular weight isoforms represented differentially glycosylated IL7R and that the 60 kD band represented the fully matured receptor. Indeed, after treatment with a general deglycosylase (PNGase), all the observed isoforms of IL7R migrated at the predicted 52 kD molecular weight (Fig. EV4A). These data suggest that Lrp10, through its ICD, either impairs IL7R glycosylation during secretion or promotes the destruction of its fully matured form.

Several variants of Lrp10 have been associated with alpha-synucleinopathies (Quadri et al, 2018). We reconstituted two of the variants, Lrp10$^{R235C}$ and Lrp10$^{R152C}$, in *Lrp10*$^{-/-}$ CD8 T cells. Lrp10$^{R235C}$ was previously shown to accumulate in enlarged vesicles in the brain of Parkinson's Disease patients, while Lrp10$^{R152C}$ did not (Grochowska et al, 2021). However, both variants were able to downregulate IL7R similarly to Lrp10$^{WT}$ (Fig. EV4B). In contrast, *Lrp10*$^{-/-}$ CD8 T cells reconstituted with the *chowmein* variant (Lrp10$^{D246Y}$) showed levels of cell surface IL7R similar to those reconstituted with the empty vector. These results suggest that the variants found in alpha-synucleinopathies do not impact Lrp10 function in our CD8 T-cell-based assay.

## Lrp10 limits CD8 T-cell tumor infiltration and anti-tumor immunity

Tumor infiltration by cytotoxic CD8 T cells correlates with improved survival and positive responses to immune checkpoint inhibition (Galon and Bruni, 2019; Lee and Ruppin, 2019; Li et al, 2021). Our data thus far showed that although *Lrp10*$^{-/-}$ mice harbored increased numbers of CD8 T cells they did not exhibit superior cytotoxic activity or recall responses after immunization with a model antigen. Therefore, we asked how *Lrp10* deletion might affect anti-tumor immune responses. We subcutaneously inoculated *Lrp10*$^{+/+}$ and *Lrp10*$^{-/-}$ mice with MC38 cells which give rise to syngeneic, highly immunogenic tumors derived from a chemically induced murine colon cancer. *Lrp10*$^{-/-}$ mice showed enhanced resistance to MC38 tumor growth (Fig. 4A,B). Immune phenotyping of infiltrating cell populations revealed that tumors

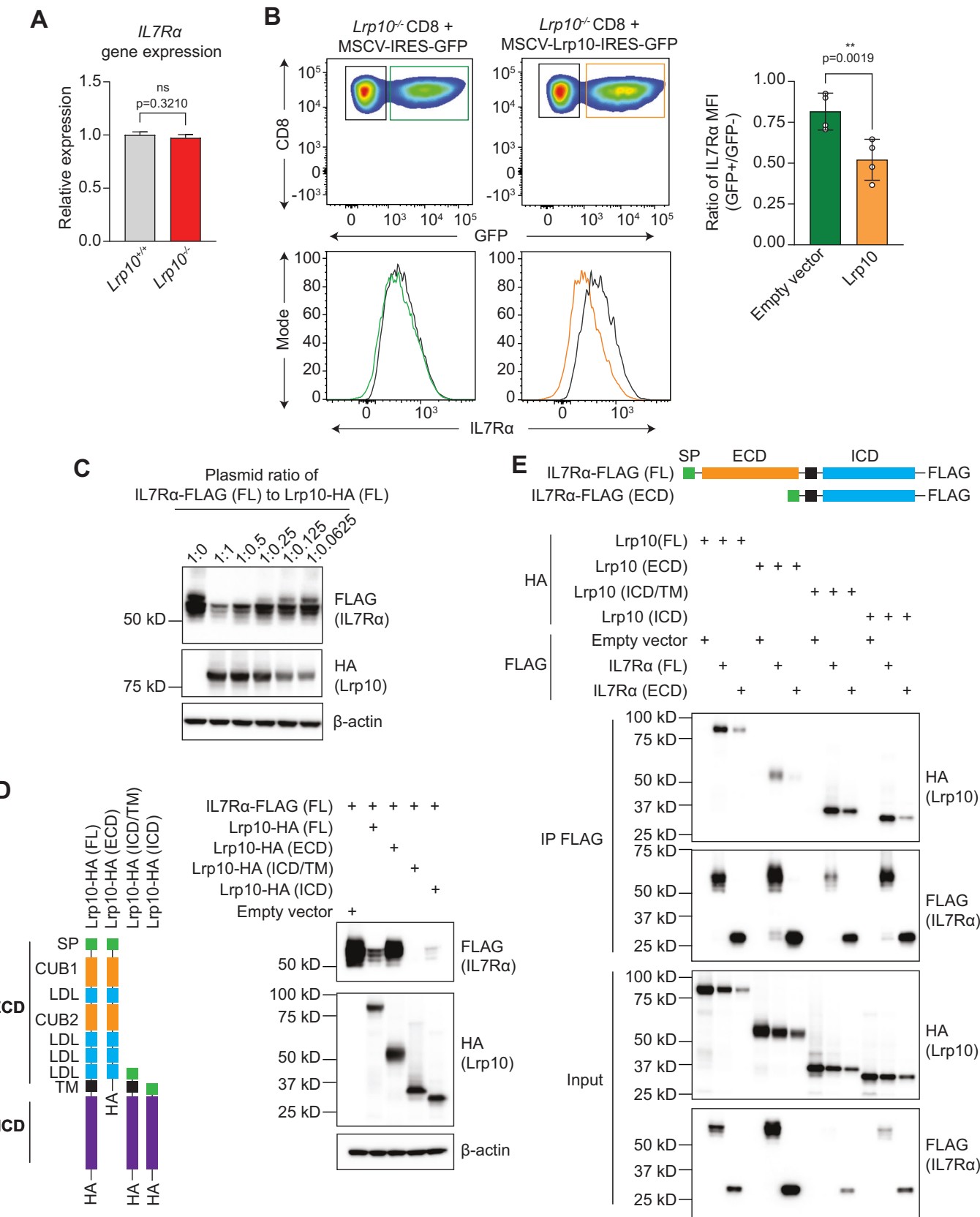

**Figure 3.  Lrp10 suppresses IL7R expression through its ICD.**

(A) RT-qPCR measurement of *IL7Ra* gene expression in splenic CD8 T cells. (B) Representative FACS plots of cell surface IL7R expression on activated *Lrp10⁻/⁻* CD8 T cells infected with retroviruses encoding GFP-only (green trace) or Lrp10^WT-IRES-GFP (yellow trace). IL7R levels on GFP+ cells were normalized to levels on the GFP-population (black trace). (C) Co-expression of full-length (FL) IL7R-FLAG with different doses of Lrp10-HA (FL) in HEK 293T cells. (D) Co-expression of IL7R-FLAG with the indicated Lrp10-HA deletion mutants HEK 293T cells. (E) Co-IP of IL7R-FLAG (FL) or IL7R-FLAG (ECD) with the indicated Lrp10-HA deletion mutants from co-transfected HEK 293T cells. Data information: In (A), horizontal bars indicate the mean value of three biological replicates per genotype and error bars show SD. (B) Symbols show the results of four separate infection experiments (biological replicates), horizontal bars indicate the mean, and error bars show SD. Western blot results (C–E) were replicated at least twice in separate transient transfection experiments. *P* values were calculated by a two-tailed unpaired *t* test (A) and two-tailed paired *t* test (B). Significant *P* values were flagged as follows: \*P < 0.05 and \*\*P < 0.01. *P* values >0.05 were considered to be not significant (ns). Source data are available online for this figure.

from *Lrp10⁻/⁻* mice harbored increased numbers of CD8 T cells, but similar numbers of CD4 T cells and macrophages (Fig. 4C). Within the CD4 population, the frequency of regulatory T cells (T_reg) was the same between each strain (Fig. EV5A). Although *Lrp10⁻/⁻* mice have fewer circulating NK cells, the numbers of NK1.1+ cells were similar in tumors from *Lrp10⁺/⁺* and *Lrp10⁻/⁻* mice (Fig. 4C). Notably, the tumor resistance phenotype in *Lrp10⁻/⁻* mice was limited to the highly immunogenic MC38 tumor. There was no significant difference in tumor growth rate, or CD8 T-cell infiltration, between *Lrp10⁺/⁺* and *Lrp10⁻/⁻* mice inoculated with the "immunologically cold" B16F10 melanoma cell line (Fig. EV5B,C).

We next determined whether the tumor resistance phenotype in *Lrp10⁻/⁻* mice depended on CD8 T cells. *Lrp10⁺/⁺* and *Lrp10⁻/⁻* were challenged with MC38 tumors and CD8 T cells were depleted in vivo through the administration of an anti-CD8 antibody (Fig. 4D). Depleting CD8 T cells from *Lrp10⁻/⁻* mice eliminated their ability to resist the MC38 tumor. Together, these data show that *Lrp10⁻/⁻* mice accumulate higher levels of CD8 T cells within immunogenic tumors which are critical for enhanced primary tumor resistance.

*Lrp10* deletion did not enhance CD8 T-cell responses against strong foreign antigens in the adoptive transfer setting. *Rag2⁻/⁻* mice were inoculated with B16 melanoma cells that constitutively expressed ovalbumin (B16-ova) and were then injected with either 10⁴ *Lrp10⁺/⁺;OT-1* or *Lrp10⁻/⁻;OT-1* cells on D6 (Fig. EV5D). Each cell population was able to induce complete eradication of B16-ova tumors with similar kinetics. While *Lrp10⁻/⁻;OT-1* cells persisted in the peripheral blood at higher levels 1 month after tumor rejection (Fig. EV5E), all mice in both treatment groups were able to resist a subsequent challenge with B16-ova cells.

## *Lrp10⁻/⁻* CD8 TILs maintain higher levels of IL7R and show reduced frequencies of terminally exhausted cells

IL7R expression on activated CD8 T cells marks cells with memory potential (Kaech et al, 2003) and is associated with enhanced responses to chronic viral infections and tumors (Belarif et al, 2018; Krishna et al, 2020; Micevic et al, 2023; Pauken et al, 2016). Given that Lrp10 suppressed IL7R during normal CD8 T-cell homeostasis, we next compared IL7R levels on CD8 T cells in tumors from *Lrp10⁺/⁺* and *Lrp10⁻/⁻* mice to those from the spleen (Fig. 4E). Splenic CD8 T cells from *Lrp10⁺/⁺* and *Lrp10⁻/⁻* mice each displayed high levels of IL7R, with elevated levels observed in *Lrp10⁻/⁻* cells. Within the tumor, *Lrp10⁺/⁺* CD8 T cells showed >50% reduction in cell surface IL7R. In contrast there was no significant reduction in IL7R levels on *Lrp10⁻/⁻* CD8 TILs.

Accordingly, tumors from *Lrp10⁻/⁻* mice showed greater frequencies of total IL7R + CD8 TILs. These data demonstrate that loss of Lrp10 allows CD8 T cells to maintain higher levels of IL7R expression in the TME.

CD8 TILs are chronically exposed to antigen within the TME, which drives their differentiation into terminally exhausted cells (Chow et al, 2022; Giles et al, 2023). In a publicly available single-cell RNAseq (scRNAseq) dataset of human CD8 T cells isolated from melanoma tumors (Sade-Feldman et al, 2018), Lrp10 gene expression overlapped with the expression of genes involved in T-cell exhaustion and was negatively correlated with genes involved in stem and memory cell function (Appendix Fig. S1). Therefore, we next asked how *Lrp10* deletion affected the phenotype of CD8 TILs during the anti-tumor immune response.

We started by measuring the expression of markers for central memory cells (CD44 and CD62L) and for terminally exhausted cells (PD1 and Tim3) (Sakuishi et al, 2010; Zhou et al, 2011). *Lrp10⁻/⁻* mice accumulated increased frequencies of T_CM phenotype cells (CD62L + CD44 + ) in MC38 tumors both at early (day 12 after inoculation) and late (day 19 after inoculation) timepoints (Fig. 4F). While tumors in *Lrp10⁺/⁺* and *Lrp10⁻/⁻* mice harbored similar frequencies of terminally exhausted (T_EX) CD8 effectors (PD1+Tim3 + ) on day 12, *Lrp10⁻/⁻* mice showed reduced frequencies of T_EX cells by day 19.

The ability to secrete inflammatory cytokines upon re-stimulation is a key characteristic of T_CM phenotype CD8 T cells. Therefore, we measured secretion of interferon-γ (IFN- γ) and tumor necrosis factor-α (TNF-α) in *Lrp10⁺/⁺* and *Lrp10⁻/⁻* CD8 TILs in vitro after re-stimulation with PMA/ionomycin (Fig. 4G). On day 12, the *Lrp10⁻/⁻* CD8 TIL population had ~2.5-fold higher frequency of cells that secreted both IFN- γ and TNF-α compared to the corresponding *Lrp10⁺/⁺* population. Importantly, on day 19, the frequency of cytokine secreting *Lrp10⁺/⁺* CD8 T cells declined ~threefold while the *Lrp10⁻/⁻* population sustained levels that were similar to the earlier timepoint. Together, these data indicate that *Lrp10* deletion skews the composition of the CD8 TIL population away from terminal exhaustion and enriches a T_CM phenotype that retains cytokine secretion capabilities.

## Single-cell transcriptional and TCR profiling define CD8 TIL heterogeneity with and without Lrp10

We next sought to better define the identity and heterogeneity of *Lrp10⁺/⁺* and *Lrp10⁻/⁻* CD8 TILs. *Lrp10⁺/⁺* and *Lrp10⁻/⁻* CD8 T cells were sorted from D12 MC38 tumors and subjected to paired scRNAseq and single-cell TCR sequencing (scTCRseq). We analyzed quality transcriptome data from 1269 *Lrp10⁺/⁺* cells and

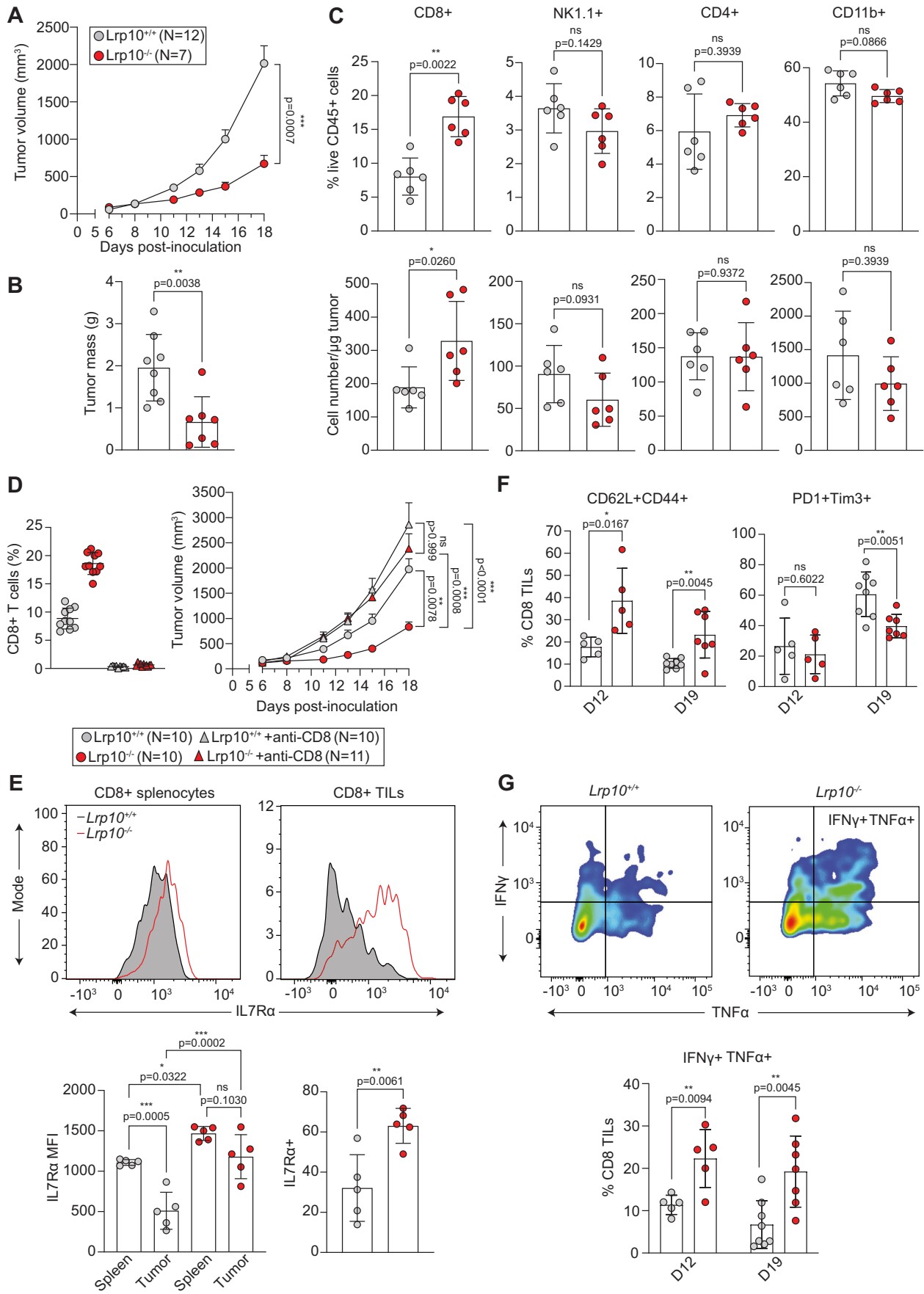

**Figure 4.** *Lrp10* deletion imparts tumor resistance.

(A) MC38 tumor volumes. Differences in tumor volume were compared on D18. (B) D18 MC38 tumor mass. (C) Quantification of CD45+ immune cells in D18 MC38 tumors. (D) The bar graph shows the efficacy of CD8 depletion. The tumor growth curve shows the effect of CD8 T-cell depletion on MC38 tumor growth. Differences in tumor volume were compared on D18. (E) FACS plot of IL7R expression on spleen and tumor CD8 T cells from D12 MC38 tumors, quantification of IL7R MFI, and frequency of IL7R + CD8 TILs. (F) Frequency of $T_{CM}$ phenotype (CD62L + CD44 + ) and $T_{EX}$ (PD1+Tim3 + ) CD8 TILs in MC38 tumors at the indicated timepoints. (G) Representative FACS plots of IFNγ and TNFα production in restimulated CD8 TILs from D12 MC38 tumors. The bar graphs show the frequencies of cytokine-producing CD8 TILs at the indicated timepoints. Data information: In bar graphs, symbols represent individual mice (biological replicates), horizontal bars indicate the mean, and error bars show SD. In tumor growth curves (A, D), symbols represent the mean and error bars show SEM. Data were replicated two (B–D, G), three (E, F), or five (A) times in experiments with separate cohorts of at least five mice. *P* values calculated by two-tailed unpaired *t* test (A, B, F, G), Mann–Whitney (C), Kruskal–Wallis test with Dunn's correction (D), or one-way ANOVA with Tukey's multiple comparisons test (E). Significant *P* values were flagged as follows: *$P < 0.05$, **$P < 0.01$, ***$P < 0.001$. *P* values > 0.05 were considered to be not significant (ns). Source data are available online for this figure.

6267 *Lrp10$^{-/-}$* cells. We performed unsupervised Louvain clustering of the pooled data, which yielded five distinct cell clusters visualized by uniform manifold approximation and projection (UMAP, Fig. 5A). Analysis of the most highly expressed genes between each cluster showed significant differences (Fig. 5B; Appendix Table S1). Cluster 0 showed high expression of naive and memory genes (*Sell, CCr7, Tcf7, Lef1, Satb1, Bach2*). Cluster 1 showed high expression of genes associated with CD8 T-cell effector function (*Gzma, Gzmb*, and *Ccl5*) and exhaustion (*Pdcd1 and Fasl*). Cluster 2 showed high expression of genes associated with innate CD8 T cells (*Klra7, Klrc2, Irak2, Fcer1g*). Cluster 3 showed high expression of genes associated with exhaustion (*Lag3, Rgs16, TNFRSF9*). Cluster 4 had very few cells of either genotype and showed high expression of the *Fos, Jun*, and *Id3* transcription factors.

While there was substantial overlap in the UMAP spaces occupied by *Lrp10$^{+/+}$* and *Lrp10$^{-/-}$* CD8 TILs, we noted key differences (Fig. 5C). *Lrp10$^{+/+}$* cells fell predominantly into cluster 1 (exhausted/effector cells) while *Lrp10$^{-/-}$* cells were found predominantly in cluster 0 (naive/memory cells). Approximately equal frequencies of *Lrp10$^{+/+}$* and *Lrp10$^{-/-}$* cells were found in clusters 2, 3, and 4.

Cells in all clusters expressed *CD44*, indicating that they had at some point undergone activation (Fig. 5D). *Sell* expression was restricted to clusters 0 and 2, while *Pdcd1* expression was restricted to clusters 1 and 3. Expression of *Tcf7*, a central regulator of CD8 T-cell memory differentiation and stem-like activity (Escobar et al, 2020; Pais Ferreira et al, 2020), was distributed between clusters 0, 1, and 2 and was largely excluded from cluster 3. Cells expressing the canonical effector genes *Gzma* and *Klrg1* were found primarily in cluster 1.

scTCRseq showed that *Lrp10$^{+/+}$* or *Lrp10$^{-/-}$* CD8 T cells that had undergone clonal expansion (>2 cells detected per clonotype) were found almost exclusively in clusters 1 (exhausted/effectors) and 3 (exhausted) which closely overlapped with *Pdcd1* expression (Fig. 5E). In contrast, clusters 0 (naive/memory cells) and 2 (innate-like cells) were predominantly composed of singlet clonotypes. The clonally expanded population in tumors from *Lrp10$^{+/+}$* mice showed a few clonotypes that each contained a large number of cells. In contrast, the clonally expanded population in tumors from *Lrp10$^{-/-}$* mice harbored a larger number of unique clonotypes that each contained relatively fewer numbers of cells (Fig. 5F). Interestingly, this finding is consistent with prior reports that enhanced IL7R signaling augments clonal diversity within responding CD8 T-cell populations and reduces immunodominance (Melchionda et al, 2005; Sportes et al, 2008).

Together, transcriptional and TCR profiling show that *Lrp10* deletion enhances the accumulation of singlet CD8 T cells that express memory cell markers within the TME, reduces the overall frequency of cells expressing exhaustion markers, and reduces clonality within the clonally expanded TIL repertoire.

## All tumor-infiltrating CD8 T cells show evidence of tumor reactivity

CD8 TILs are a heterogeneous population comprised of both tumor antigen-specific clones and bystander cells (Meier et al, 2022; Simoni et al, 2018). While bystander cells are associated with enhanced anti-tumor responses, the reasons for their accumulation and the mechanisms through which they act to influence tumor immunity are not well understood. The memory phenotype population that accumulated in tumors from *Lrp10$^{-/-}$* mice did not show evidence of clonal expansion, therefore we speculated that they might be bystander cells.

To help identify tumor-reactive CD8 T cells within MC38 tumors, we crossed *Lrp10$^{-/-}$* mice to the *Nur77$^{GFP}$* reporter strain, which exhibits GFP expression proportional to TCR stimulation (Au-Yeung et al, 2014; Moran et al, 2011). GFP expression in CD8 T-cell subsets from naive *Lrp10$^{-/-}$;Nur77$^{GFP}$* mice was low and was not substantially different compared to cells from *Lrp10$^{+/+}$;Nur77$^{GFP}$* mice (Fig. EV5F). We challenged *Lrp10$^{+/+}$;Nur77$^{GFP}$* and *Lrp10$^{-/-}$;Nur77$^{GFP}$* mice with subcutaneous MC38 cells and harvested spleens and tumors on day 18 post-inoculation (Fig. 5G). GFP signal was low in splenic CD8 $T_{CM}$ cells from tumor-bearing mice and there was no Lrp10-dependent difference in GFP intensity. Compared to the splenic populations, GFP expression was increased in CD8 TILs from both *Lrp10$^{+/+}$;Nur77$^{GFP}$* and *Lrp10$^{-/-}$;Nur77$^{GFP}$* mice, indicating heightened tumor reactivity. GFP signal from CD8 + $T_{CM}$ and $T_{EX}$ cells were similar in tumors from *Lrp10$^{+/+}$;Nur77$^{GFP}$* mice. In contrast, $T_{EX}$ cells from the *Lrp10$^{-/-}$;Nur77$^{GFP}$* TIL population displayed increased levels of GFP signal compared to $T_{CM}$ cells, suggesting that terminally differentiated *Lrp10$^{-/-}$* CD8 TILs experienced higher levels of TCR signaling.

Together, these data show that the majority of CD8 T-cell-infiltrating tumors have some degree of tumor reactivity regardless of Lrp10 status. Significantly, despite not undergoing clonal expansion, central memory phenotype CD8 TILs from *Lrp10$^{-/-}$* mice showed evidence of active TCR signaling and thus were not tumor-ignorant, inactive bystanders. We found that CD8 TILs with a $T_{CM}$ phenotype expressed significantly higher cell surface levels of CD5 which corresponded with our findings in splenic CD8 T cells

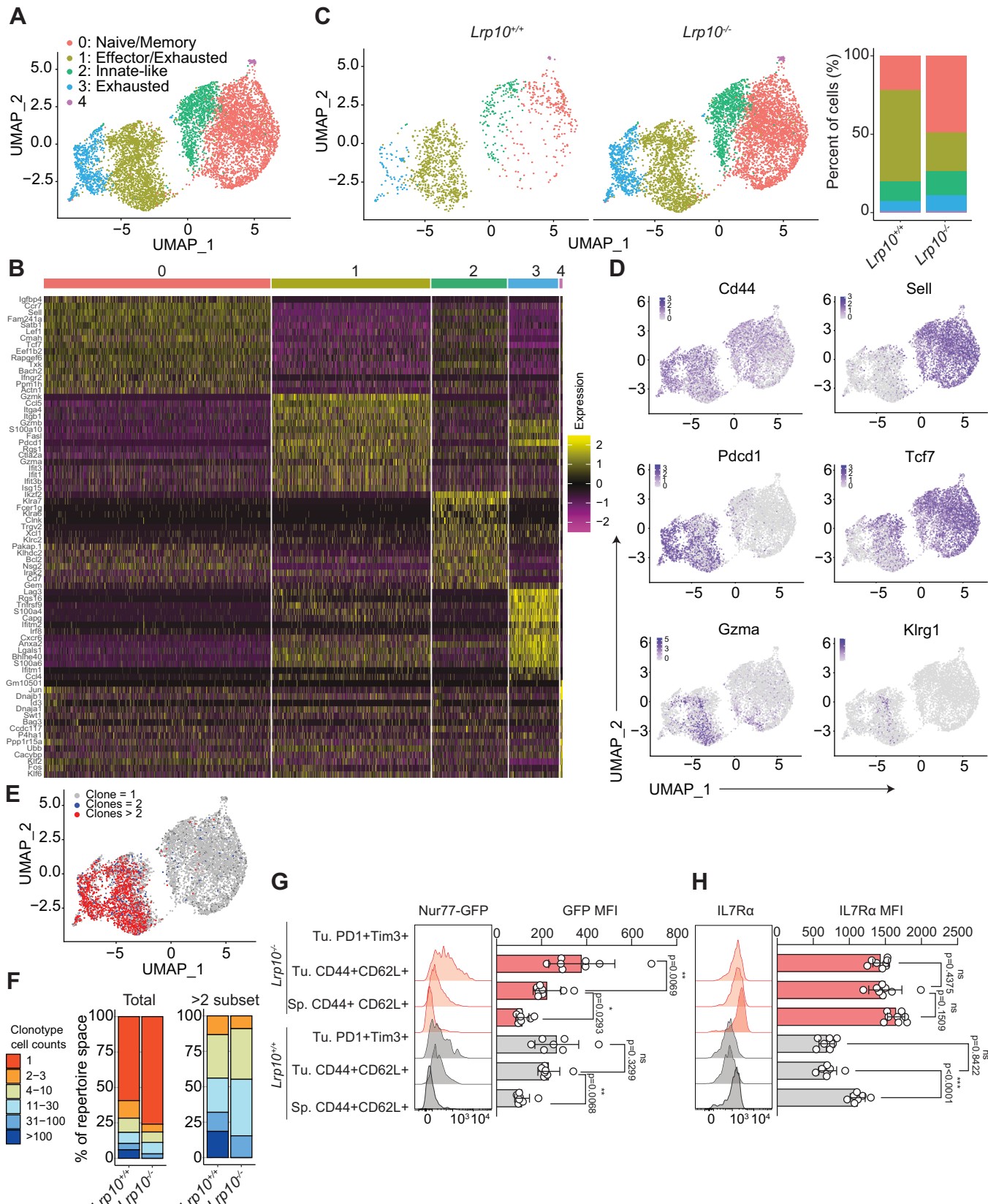

**Figure 5.** *Lrp10* deletion increases bystander CD8 T-cell accumulation.

(A) UMAP of merged scRNAseq data from 1269 *Lrp10*[+/+] cells and 6267 *Lrp10*[−/−] CD8 TILs sorted from D12 MC38 tumors. Each dot corresponds to one individual cell. A total of 5 clusters (cluster 0 through 4) were identified and color-coded. (B) A heatmap of the 15 most highly expressed genes in each cluster from (A). Columns correspond to individual cells and rows correspond to genes. Color scale is derived from the z-score distribution from -2 (purple) to 2 (yellow). (C) Contribution of *Lrp10*[+/+] and *Lrp10*[−/−] cells to the UMAP clusters identified in (A). (D) Distribution of single-cell transcript levels for *Cd44, Sell, Pdcd1, Tcf7, Gzma*, and *Klrg1* in the UMAP from (A). Purple indicates gene expression and gray indicates no expression. (E) Overlay of TCR clonotypes containing >2 cells, =2 cells, or =1 cell on the UMAP from (A). (F) Stacked bar graph of relative clonotype abundance within the total CD8 TIL population and within the subset of clonotypes containing >2 cells. (G, H) Representative FACS histograms of GFP and IL7R expression in splenic CD8 T$_{CM}$ cells and CD8 TIL subpopulations from mice with D18 MC38 tumors. Bar graphs show GFP and IL7R MFI. Data information: In bar graphs (G, H), symbols represent individual mice (biological replicates), horizontal bars indicate the mean, and error bars show SD. Data in (G, H) were replicated on two separate cohorts of at least five mice. *P* values were calculated using one-way ANOVA with Holm–Sidak's test. Significant *P* values were flagged as follows: *$P < 0.05$, **$P < 0.01$, ***$P < 0.001$. *P* values > 0.05 were considered to be not significant (ns). Source data are available online for this figure.

from unchallenged mice and suggest heightened self-reactivity (Fig. EV5G). Therefore, we suspect that T$_{CM}$ phenotype TILs are cross-reactive between endogenous/self-antigens and tumor antigens and that *Lrp10* deletion allows these cells to persist within the CD8 repertoire.

TCR signaling downregulates IL7R expression (Chandele et al, 2008). Accordingly, GFP + CD8 TILs from *Lrp10*[+/+]*;Nur77*[eGFP] mice showed a decline in cell surface expression of IL7R compared to splenic T$_{CM}$ cells (Fig. 5H). Notably, CD8 TILs from *Lrp10*[−/−]*;Nur77*[eGFP] mice showed sustained IL7R expression, even in terminally differentiated T$_{EX}$ cells, suggesting that *Lrp10* deletion uncouples IL7R expression from chronic TCR signaling.

## *Lrp10* deletion skews the clonally expanded population away from exhaustion

We next assessed how *Lrp10* deletion affected the differentiation of clonally expanded CD8 T cells in MC38 tumors. We analyzed differential gene expression between *Lrp10*[+/+] and *Lrp10*[−/−] CD8 TILs for clonotypes that contained more than two cells, corresponding to a subpopulation of cells within clusters 1 and 3 in the original UMAP (Fig. 6A; Appendix Table S2). Among genes upregulated in clonally expanded *Lrp10*[−/−] CD8 TILs were those commonly associated with interferon responses (*Ifitm1, Ifitm2, Ifitm3, Plac8, Capg*) and cytotoxicity (*Jaml, Ctsw, Klrk1*). In contrast, *Lrp10*[+/+] clonally expanded cells showed increased expression of epigenetic regulators associated with T-cell exhaustion (*Tox*) and hematopoietic differentiation (*Jarid2*) (Kinkel et al, 2015) as well as *Fgfr2* and *Hexb*. We did not note any difference in the expression of stem-memory genes (*IL7R, Tcf7*) or in other transcriptional regulators of exhaustion (*Tbx21, Eomes, Prdm1*) (Fig. 6B).

Given the importance of *Tox* in CD8 T-cell exhaustion (Khan et al, 2019; Scott et al, 2019; Seo et al, 2019; Yao et al, 2019), and its apparent upregulation in clonally expanded *Lrp10*[+/+] CD8 TILs, we specifically measured Tox protein expression in CD8 TILs in day 12 MC38 tumors (Fig. 6C,D). We used PD1 expression as a surrogate marker for clonally expanded CD8 TILs given the high degree of overlap between *Pdcd1* expression and clonotypes containing two or more cells. Within the PD1+ subpopulation, *Lrp10*[−/−] cells CD8 TILs displayed lower levels of *Tox* expression, consistent with our scRNAseq analysis. Comparing IL7R and Tox levels between PD1+ *Lrp10*[+/+] and *Lrp10*[−/−] CD8 TILs showed that the *Lrp10*[−/−] population was skewed toward increased IL7R expression and decreased Tox expression whereas this relationship was inverted in *Lrp10*[+/+] cells. These data support the conclusion that *Lrp10*

deletion attenuates CD8 TIL terminal exhaustion in tumors and suggests that Lrp10 potentiates Tox-induced exhaustion programming.

Reclustering the subset of clonotypes containing 2 or more cells resulted in a UMAP partitioned into four subclusters (Fig. 6E). Analysis of the differentially expressed genes between each cluster showed high expression of *Tcf7* in cluster 0, together with effector genes like *Ccl5* and *Gzma* (Fig. 6F; Appendix Table S3). This profile suggests that cluster 0 represents stem/memory anti-tumor CD8 T cells. Cluster 1 showed high expression of genes associated with exhaustion (*Lag3, Rgs16, Tnfrsf9*) and effector function (*Irf8, Tnfrsf4, Il2ra, Ccl4*), indicating an intermediate effector/exhausted phenotype. In contrast, cluster 2 showed upregulation of multiple factors known to facilitate terminal exhaustion (*Tox, Ikzf2, Maf*). Cluster 3 showed upregulation of cell cycle genes (*MKi67, Pclaf, Stmn1, Tubb5, Hmgb2*) indicating a subpopulation of proliferative cells.

Most *Lrp10*[+/+] and *Lrp10*[−/−] cells were in cluster 0 (stem/memory) where they occurred at approximately equal frequencies (Fig. 6G). *Lrp10*[−/−] cells also partitioned into clusters 1 (intermediate effectors/exhausted) and 3 (cycling). In contrast, after cluster 0, most *Lrp10*[+/+] cells were found in cluster 2 (terminally exhausted). *Lrp10*[+/+] cells were found to a lesser extent in cluster 1 and were absent from cluster 3. Together, these data suggest that *Lrp10* deletion reduces the frequency of terminal exhaustion amongst clonally expanded CD8 TILs and, instead, enriches cells with an intermediate effector/exhausted phenotype or a proliferative phenotype.

## *Lrp10* deletion synergizes with anti-PD1 immunotherapy to cure tumors at high frequencies

The frequency of CD8 terminal exhaustion in the TME is inversely correlated with the efficacy of immune checkpoint inhibition in chronic viral infections and cancer (Kurtulus et al, 2019; McLane et al, 2019; Miller et al, 2019; Sade-Feldman et al, 2018). The immunological and transcriptional phenotype of *Lrp10*[−/−] CD8 TILs suggested that they were resistant to terminal exhaustion and instead primarily expressed effector or memory features. Therefore, we hypothesized that *Lrp10* deletion would enhance the effect of immune checkpoint inhibition. We injected *Lrp10*[+/+] and *Lrp10*[−/−] mice with 2.5 mg/kg of anti-PD1 for three doses starting on day 6 after inoculation with MC38 cells (Fig. 6H). Anti-PD1 at this dose imparted partial MC38 resistance to *Lrp10*[+/+] mice which resembled the tumor growth rate in *Lrp10*[−/−] mice treated with vehicle. In contrast, anti-PD1 treatment strongly synergized with

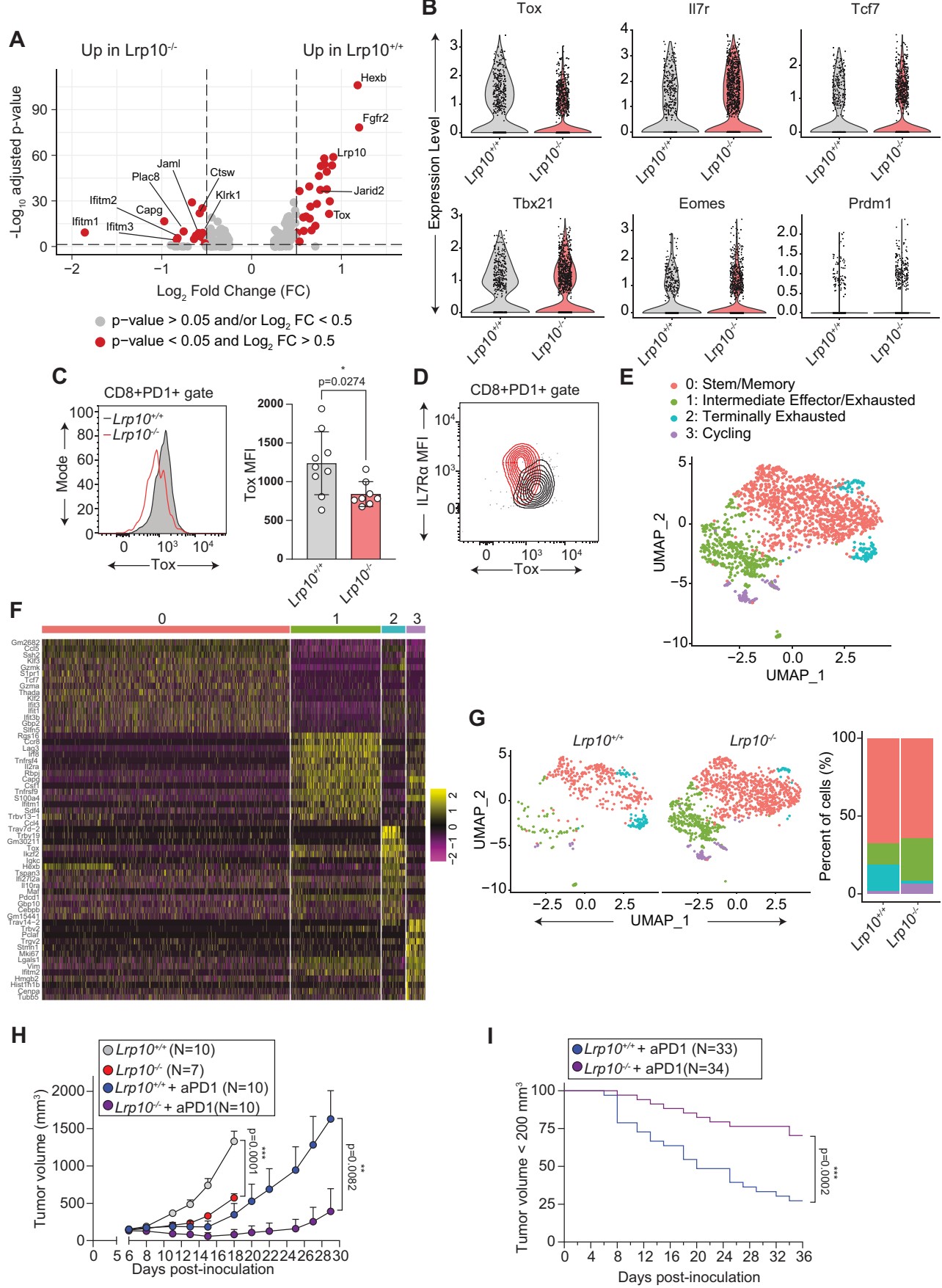

◄ **Figure 6. *Lrp10* deletion counteracts exhaustion programming in clonally expanded CD8 TILs.**

(A) Volcano plot showing differentially expressed genes in *Lrp10*[+/+] and *Lrp10*[−/−] CD8 TIL clonotypes containing >2 cells. (B) Violin plots of *IL7R, Tox, Tcf7, Fgfr2, Tbx21, Eomes, Prdm1*. Symbols represent individual cells, $n = 467$ (*Lrp10*[+/+]) and 1206 (*Lrp10*[−/−]). (C) Representative FACS histogram of Tox expression and quantification of Tox expression in D12 CD8 + PD1+ TILs. (D) Representative FACS plot of Tox vs IL7R expression in CD8 + PD1+ TILs from the mice in (C). (E) UMAP of reclustered scRNAseq data from 1673 CD8 TILs containing clonotypes with more than 2 cells. Each dot corresponds to one individual cell. A total of 4 clusters (cluster 0 through 3) were identified and color-coded. (F) A heatmap of the 15 most highly expressed genes in each cluster from panel E. Columns correspond to individual cells and rows correspond to genes. Color scale is derived from the z-score distribution from −2 (purple) to 2 (yellow). (G) Contribution of *Lrp10*[+/+] and *Lrp10*[−/−] cells to the UMAP clusters identified in (E). (H) MC38 tumor growth in *Lrp10*[+/+] and *Lrp10*[−/−] mice with and without anti-PD1. (I) Frequency of MC38 tumor progression in *Lrp10*[+/+] and *Lrp10*[−/−] mice treated with anti-PD1 across three separate cohorts. Mice were said to have progressed if tumor volume exceeded 200 mm³. Data information: In (A), statistical analysis was performed using a nonparametric Wilcoxon rank sum test with $n = 467$ (*Lrp10*[+/+]) and 1206 (*Lrp10*[−/−]) cells. (C) Symbols represent individual mice (biological replicates), horizontal bars indicate mean values, and error bars show SD. The data shown in (C) is combined from two separate experiments. (H) Symbols represent the mean value and error bars show SEM. Results in (H) were replicated in three separate cohorts of at least eight mice. *P* values were calculated by two-tailed unpaired *t* test (C), Mann–Whitney (H), and log-rank (Mantel–Cox) test (I). Significant *P* values were flagged as follows: *$P < 0.05$, **$P < 0.01$, ***$P < 0.001$. *P* values >0.05 were considered to be not significant (ns). Source data are available online for this figure.

*Lrp10* deletion to slow the growth of MC38 tumors. Over several cohorts of mice, we found that anti-PD1 treatment coupled with *Lrp10* deletion resulted in improved survival compared to anti-PD1 treatment alone (Fig. 6I). Tumor immunogenicity was important for synergy with immune checkpoint inhibition: anti-PD1 combined with *Lrp10* deletion did not have a major effect in the B16F10 model where we had observed no primary effect on tumor growth or CD8 T-cell infiltration with *Lrp10* deletion alone (Fig. EV5H). Overall, *Lrp10* deletion imparts both partial intrinsic resistance to immunogenic tumors and enhances the susceptibility of these tumors to immune checkpoint inhibition.

## Discussion

Through forward genetic screening in mice, we discovered that *Lrp10* maintains CD8 T-cell homeostasis by limiting the size of the T$_N$ and T$_{CM}$ subpopulations. Our data suggest a model where Lrp10 downregulates IL7R to curtail the number of CD8 T cells that survive in peripheral lymphoid organs. We further propose that induction of Lrp10 during TCR stimulation reduces the number of CD8 T cells able to compete for IL7 and enter the central memory repertoire. We hypothesize that Lrp10 may help drive CD8 T-cell responses toward immunodominant antigens by limiting the number and diversity of cells that persist after clonal contraction.

Our study of Lrp10 presents a new component of IL7R regulation. One possible reason for this type of regulation is to prevent the propagation of long-lived memory cells expressing potentially self-reactive antigen receptors. Notably, enhanced IL7-IL7R signaling has been linked to multiple autoimmune diseases (Lundstrom et al, 2012). We found that T$_{CM}$ and T$_{EM}$ CD8 T cells from *Lrp10*[−/−] mice displayed increased expression of CD5, a negative regulator of TCR signaling that is associated with self-reactive T cells. However, *Lrp10*[−/−] mice did not show outward manifestations of spontaneous autoimmunity like arthritis, dermatitis, lymphoproliferative disease, or decreased survival that is seen with deletion of other T-cell-negative regulators (Perry et al, 1998; Waterhouse et al, 1995). C57BL/6 mice are generally resistant to spontaneous autoimmunity and usually require immunization to break tolerance. Given the cellular and molecular phenotypes of *Lrp10*[−/−] mice, it will be interesting to test whether they have increased susceptibility to induced autoimmunity and whether

*Lrp10* deletion would worsen spontaneous autoimmunity on sensitized genetic backgrounds like NOD (Gearty et al, 2022). In addition, *Lrp10*[−/−] mice are currently housed under specific pathogen-free conditions in which the environmental antigen burden is low (Beura et al, 2016). Whether exposing *Lrp10*[−/−] mice to "dirtier" environments would amplify cross-reactive T$_{CM}$ populations with the potential to cause autoimmune tissue damage remains to be tested.

Another factor that might contribute to the lack of overt spontaneous autoimmunity in *Lrp10*[−/−] mice is that CD4 T cells are not affected as much as CD8 T cells. We have shown that Lrp10 is induced with T-cell activation and several lines of evidence have previously shown that activation of CD8 versus CD4 T cells is quantifiably different: CD8 T cells have a lower threshold for activation whereas CD4 T cells have an increased requirement for co-stimulatory molecules and specific cytokines (Seder and Ahmed, 2003). Furthermore, we showed that TCR restriction with OT-1 reduces the frequency of CD8 T$_{CM}$ cells in *Lrp10*[−/−] mice and suggests that at some point TCR-MHC-I interactions promote their accumulation. Most cells in the body express MHC-I, thus providing many opportunities for interactions with CD8 T cells, while MHC-II is restricted to specific subsets of cells. If CD4 T cells have fewer opportunities for TCR-MHC-II interactions, and are intrinsically more resistant to stimulation, it may explain why *Lrp10* deletion does not have as large an effect on expanding the CD4 T-cell compartment. Overall, better understanding of the factors that control Lrp10 expression and function, such as TCR signal strength, co-stimulation, and cytokine milieu, will provide important insights into how Lrp10 shapes CD8 versus CD4 T-cell fate.

Deletion of *Lrp10* also imparts intrinsic resistance to immunogenic tumors. This phenotype depends on extensive infiltration of CD8 T-cell populations that harbor reduced frequencies of T$_{EX}$ cells, retain high levels of IL7R and other memory cell characteristics, and display enhanced susceptibility to immune checkpoint inhibition. Our data suggest two ways in which *Lrp10* deficiency influences anti-tumor immunity. First, it impairs terminal exhaustion of clonally expanded anti-tumor CD8 T cells. We showed that lack of Lrp10 suppresses the frequency cells expressing *Tox*. The mechanism that underpins this finding remains unclear. We hypothesize that elevated IL7R expression on *Lrp10*[−/−] CD8 T cells helps them resist terminal differentiation. One possibility is that enhanced IL7R signaling directly counteracts

Tox, for example, through activated STAT5 signaling networks (Ding et al, 2020). In this scenario, rational combination of IL7R agonists with immune checkpoint inhibition may benefit immunotherapy approaches. Our data also suggest that Lrp10 acts as a brake on IL7R signaling and thus may limit IL7-based therapies. Whether the ability of *Lrp10* deletion to impair T-cell exhaustion depends exclusively on IL7R, or whether other signaling networks are involved, remains to be tested.

A second way *Lrp10* deletion may promote anti-tumor immunity is through the accumulation of numerous singlet CD8 clones with a $T_{CM}$ phenotype within the TME. Based on their lack of clonal amplification and absent PD1 expression, these cells appeared to be tumor-ignorant bystanders. However, they show TCR signaling above the background, indicating that they are tumor-reactive. One possibility is that these CD8 T cells express TCRs that are cross-reactive with tumor antigens and endogenous, self, or microbial antigens, a population called "false bystanders" (Bessell et al, 2020; Chiou et al, 2021; Meier et al, 2022). The elevated CD5 expression that we observe in CD8 $T_{CM}$ phenotype TILs may indicate that they are cross-reactive with self-antigens. We suspect *Lrp10*$^{-/-}$ mice inherently possess higher frequencies of these cells due to upregulated IL7R which permits CD8 T cells with cross-reactive TCRs to persist in the central memory pool. Cross-reactive CD8 T cells are frequently found in human tumors and mouse cancer models (Caushi et al, 2021; Danahy et al, 2020; Simoni et al, 2018). There is clinical interest in exploiting these cells for functional anti-tumor responses (Batich et al, 2017; Millar et al, 2020; Rosato et al, 2019). Our data suggest *Lrp10* is part of a previously unrecognized genetic program that limits the number of false bystanders in tumors.

We have shown that Lrp10 suppresses IL7R through the Lrp10-ICD. Additionally, we noted that Lrp10 prevents the expression of the fully glycosylated form of IL7R. This may occur through two possible mechanisms: (1) Lrp10 interferes with IL7R maturation in the Golgi (for example, by blocking access to protein glycosyltransferases), or (2) Lrp10 induces the destruction of mature IL7R. Interestingly, the Lrp10-ICD is proline-rich and contains a PPxY motif. These motifs bind to WW domains found in HECT-family NEDD4-like E3 ubiquitin ligases to augment ubiquitin-ligase activity (Riling et al, 2015). The Lrp10 relative LRAD3 was previously shown to activate the NEDD4-like E3 ligase Itch through its two PPxY motifs (Noyes et al, 2016). Whether the PPxY motif of Lrp10 activates NEDD4-like E3 ligases to target IL7R for ubiquitination remains to be determined.

Prior studies have also shown that Lrp10 traffics between the trans-Golgi, plasma membrane, and endosomes (Boucher et al, 2008; Doray et al, 2008). It is currently unclear where in its membrane trafficking itinerary Lrp10 interferes with IL7R. Two DXXLL motifs in the Lrp10-ICD bind to the AP1/AP2 and GGA1 membrane trafficking complexes to mediate receptor transport from the plasma membrane to endosomes, and from endosomes to the trans-Golgi (Boucher et al, 2008; Brodeur et al, 2012). Mutation of these motifs were shown to increase retention of Lrp10 at the cell surface and within early endosomes. It is possible that blocking endocytosis of Lrp10 through DXXL mutation may increase IL7R expression, suggesting that Lrp10 facilitates IL7R removal from the cell surface and promotes its delivery to the endo/lysosomal system. Furthermore, the discovery of ligands, or co-receptors, that control Lrp10 function could reveal specific environmental cues that guide the differentiation and fate of activated T cells through IL7R signaling.

The number of NK cells was reduced in *Lrp10*$^{-/-}$ mice with a corresponding deficit in in vivo NK cytolytic activity. The reason for the NK cell deficiency is currently unknown. It is possible that Lrp10 may have a cell-intrinsic role in the development, or maintenance, of the NK cell lineage. Alternatively, CD8 T cells may modulate NK cell homeostasis by interfering with their access to key survival factors like IL15. Specific deletion of *Lrp10* in T cells versus NK cells would help define its role in NK cell homeostasis.

Through adoptive transfer studies, we demonstrated that Lrp10 acts in a cell-autonomous manner to limit IL7R expression and suppress CD8 T-cell homeostatic expansion. One significant limitation of the current study is the use of a constitutive knockout model to define Lrp10's function in anti-tumor immune responses. While this approach can approximate what happens during global inhibition of Lrp10 in a therapeutic setting, it does not rule out that loss of Lrp10 in other cell populations (e.g., stromal, myeloid, or CD4 T cells) may influence CD8 T-cell function. In addition, although *Lrp10* deletion did not distort thymic cellularity, it may change thymic T-cell developmental trajectories, or TCR selection criteria, to enhance tumor resistance. Specifying the lineages and timeframe in which *Lrp10* is deleted would help further define its role in anti-tumor immunity.

Another limitation of this study is our finding that the tumor resistance phenotype of *Lrp10*$^{-/-}$ mice was restricted to the highly immunogenic MC38 tumor. Currently, it is not clear what tumor factors dictate immune responsiveness in the setting of *Lrp10* deletion. We hypothesize that a high tumor mutational burden is important for generating the increased polyclonality observed amongst *Lrp10*$^{-/-}$ CD8 TILs. Alternatively, certain tumor types may express specific ligands for Lrp10 that increase its immunomodulatory activity. Increased expression of Lrp10 in human hepatocellular carcinoma, lung adenocarcinoma, and pancreatic adenocarcinoma is associated with decreased patient survival (Gonias et al, 2017). Whether Lrp10 expression in these scenarios derives from CD8 T cells or other cell populations within tumors is unknown. Future studies that define how tumor type and tumor mutational burden influence the clonality and differentiation phenotypes of CD8 TILs in the context of *Lrp10* deletion will be informative.

In summary, we present Lrp10 as a new determinant of IL7R expression in T cells that has important implications for CD8 T-cell fate decisions during normal homeostasis and anti-tumor immune responses.

## Methods

### Mouse strains

Mice were housed in specific pathogen-free conditions and fed a normal chow diet at the University of Texas Southwestern Medical Center. All animal experiments were performed according to institutionally approved protocols. ENU-mutagenesis, strategic breeding of mutagenized mice, phenotypic screening, and automated meiotic mapping were performed as previously described (Wang et al, 2015). B6.SJL-Ptprca Pepcb/BoyJ (*CD45.1*), C57BL/6-Tg(Nr4a1-EGFP/cre)820Khog/J (*Nur77*$^{GFP}$), C57BL/6-Tg(TcraTcrb)1100Mjb/J (*OT-1*), B6.129P2-B2mtm1Unc/DcrJ

($B2m^{-/-}$), and B6.Cg-Rag2tm1.1Cgn/J ($Rag2^{-/-}$) strains were obtained from the Jackson Laboratory. These strains were intercrossed with $Lrp10^{-/-}$ mice as needed. Male and female mice aged 8–16 weeks were used for experiments.

## Generation of knockout mouse strains using the CRISPR/Cas9 system

To generate single knockout mouse strains, female C57BL/6J mice were super-ovulated by injection of 6.5 U pregnant mare serum gonadotropin (PMSG; Millipore), followed by injection of 6.5 U human chorionic gonadotropin (hCG; Sigma-Aldrich) 48 h later. The super-ovulated mice were subsequently mated overnight with C57BL/6J male mice. The following day, fertilized eggs were collected from the oviducts. In vitro-transcribed Cas9 mRNA (50 ng/µl) and $Lrp10$ small base-pairing guide RNA (50 ng/µl) were injected into the cytoplasm or pronucleus of embryos. Injected embryos were cultured in M16 medium (Sigma-Aldrich) at 37 °C in 5% $CO_2$. For the production of mutant mice, two-cell stage embryos were transferred into the ampulla of the oviduct (10–20 embryos per oviduct) of pseudo-pregnant Hsd:ICR (CD-1) female mice (Harlan Laboratories). An $Lrp10$ CRISPR allele that resulted in a frame-shifting 8 bp deletion in exon 5 was used for all experiments. $Lrp10$ deletion was verified at the protein level through western blotting.

## Plasmids

Full-length mouse Lrp10 and Lrp10-ECD were tagged with a C-terminal HA epitope in pCMV3 for use in heterologous expression and co-immunoprecipitation experiments. For retro-viral complementation experiments, full-length Lrp10 was sub-cloned into MSCV-IRES-GFP. Mouse CD127 with a C-terminal FLAG tag in pCMV3 was obtained from SinoBiological. Point mutations and deletions were generated with the Q5 Site-Directed Mutagenesis Kit (New England Biolabs) and specific primers. Details of plasmids are available upon request. MSCV-IRES-GFP was a gift from Tannishtha Reya, Addgene plasmid # 20672.

## Adoptive transfer experiments and in vivo CD8 T-cell functional assays

For homeostatic proliferation experiments, recipient mice were sub-lethally irradiated with 6 Gy given as a single dose 24 h prior to cell transfer (X-RAD 320, Precision X-ray). Naive CD8 T cells were purified to >90% from the spleens of donor strains using negative selection (StemCell Technologies) and stained with CTV or CTFR proliferation dyes (Molecular Probes) according to the manufac-turer's instructions. Labeled cells were combined at a 1:1 ratio and 1–2 million cells were injected intravenously into recipients. For IL7R blockade, recipient mice were given intraperitoneal injections of 500 µg anti-IL7R (clone A7R34, BioXCell) or isotype control (clone 2A3, BioXCell) on days 1, 3, and 5 post-cell transfer. In immunization experiments, unirradiated mice were given a single intraperitoneal injection of 100 µg SIINFEKL (Vivitide) or SIITFEKL (ANASPEC) 24 h after cell transfer. Spleens were harvested at the indicated timepoints for analysis by flow cytometry. The frequency and proliferation status of donor cells were measured based on the indicated markers and the dilution of

the proliferation dyes. In vivo CD8 and NK cytotoxicity assays were performed as previously described (Choi et al, 2019).

## Tumor models

C57Bl/6 syngeneic MC38 (ATCC) and B16F10 (ATCC) cells were maintained in DMEM with 10% FBS and passaged three times per week. Cells were discarded after 30 passages. Mice 8–16 weeks of age were injected subcutaneously with $5 \times 10^5$ (MC38) or $4 \times 10^5$ (B16F10) cells on the right flank. Approximately equal numbers of male and female mice of each genotype were used in experiments. Tumor volume was measured three times weekly starting on day 6 post-inoculation and was calculated based on the formula length × width × width/2. For immune checkpoint inhibition experiments, mice were given intraperitoneal injections of anti-PD1 (clone RMP1-14, BioXCell) at the indicated dosages on days 5, 8, and 11 post-tumor inoculation. For CD8 depletion experiments, mice were given intraperitoneal injections of 10 mg/kg anti-CD8 (clone YTS 169.4, BioXCell).

## CD8 TIL isolation and flow cytometry

At the indicated timepoints, tumors were harvested, minced with razor blades in RPMI, and digested using the Miltenyi Tumor Dissociation Kit for 1 h at 37 °C according to the manufacturer's protocol, except for reducing the amount of enzyme R by 90% to preserve cell surface epitopes. Dissociated tumors were passed through 70-µm filters, and the volume of tumor cell suspension equivalent to 150 µg of tumor was stained with cell viability dye and the indicated cell surface markers. For cytokine production analyses, the volume of tumor cell suspension equivalent to 150 µg of the tumor was stimulated with a cocktail of PMA, ionomycin, and Brefeldin A for 4 h at 37 °C followed by staining live-cell dye and antibodies against cell surface markers. Cells were then fixed and permeabilized with the Intracellular Fixation & Permeabilization Buffer Set (eBioscience) and then stained with fluorochrome-conjugated anti-cytokine antibodies. The Mouse FoxP3 Fixation and Permeabilization Buffer Set (BDBiosciences) was used for Tox intracellular staining. Flow cytometry data was collected on an LSR Fortessa (BDBiosciences) and analyzed using FlowJo software.

## Transfection, co-immunoprecipitation assays, and western blotting

HEK 293T cells were maintained in DMEM containing 10% FBS and routinely tested for mycoplasma (Fisher Scientific). Cells were transfected in six-well plates with 1 µg of the indicated constructs, unless otherwise noted in the text or figure, and the PolyJet DNA transfection reagent (SignaGen) according to the manufacturer's instructions. At 48 h post-transfection, cells were rinsed in cold PBS and lysed in buffer containing 1% NP-40 and HALT protease inhibitor (Thermo) followed by centrifugation at 13,000 rcf for 10 min. Co-IP of FLAG-tagged proteins was performed by incubating M2 anti-FLAG resin (Sigma) with clarified cell lysates for 2 h at 4 °C with end-over-end rotation. Beads were washed four times in cold lysis buffer, and protein complexes were eluted with 150 mg/ml of 3× FLAG peptide (Sigma). Samples were diluted in 4X SDS sample buffer and analyzed with SDS-PAGE according to standard procedures. For analysis of differential IL7Ra glycosylation, transfected cells were lysed, sonicated,

and boiled in 200 μL of buffer containing 1% SDS, HALT protease inhibitor, and benzonase (Sigma). Denatured lysates were then diluted to 1.5 mL in 1% NP-40 and anti-FLAG IP was performed as usual. Eluted proteins were then treated with PNGase-F (Promega) according to the manufacturer's protocol and analyzed with SDS-PAGE. For western blotting on primary cells, cell pellets were lysed in buffer containing 1% SDS, HALT protease inhibitor, and benzonase. Protein levels were normalized using the bicinchoninic acid (BCA) assay (Pierce) and 10–15 μg of protein was diluted in 4× SDS sample buffer and analyzed with SDS-PAGE.

## Retroviral production

HEK 293T cells were transfected with 10 μg the specified pMSCV plasmid and 5 μg of pCL-Eco plasmid mixec with 45 μg of Polyethylenimine (PEI) resuspended in OptiMEM. Viral supernatants were collected between 48 and 72 h and concentrated with Retro-X concentrator (Takara) according to the manufacturer's instructions. Concentrated MSCV retroviral supernatants were used immediately for T-cell infections.

## Reconstitution of $Lrp10^{-/-}$ CD8 T cells

Splenocytes from $Lrp10^{-/-};OT-1$ mice were stimulated at 1 million cells/ml with 1 μM of SIINFEKL in 12-well plates. 24 h later, stimulated T cells were infected with 100 μL of concentrated MSCV retroviral supernatants and spin-infected for 1 h at 1800 rpm. Cells were harvested for analysis 48 h after infection. An uninfected population of stimulated $Lrp10^{-/-};OT-1$ was spiked into the infected populations and cells were stained for CD8 and IL7R and analyzed by flow cytometry. Downregulation of IL7R was assessed by dividing the IL7R gMFI in the GFP-positive population by the IL7R gMFI in the GFP-negative population.

## Real-time quantitative PCR measurements

RNA from purified splenic CD8 T cells was reverse transcribed into cDNA using oligo-d(T) primers and M-MuLV reverse transcription (Promega). Real-time quantitative PCR was performed using SYBR DNA polymerase (Thermo Scientific) and target-specific primers.

## scRNAseq and scTCRseq

In total, $1.5 \times 10^4$ live CD8 T cells were FACS sorted from dissociated D12 MC38 tumors. scRNAseq and scTCRseq libraries were generated using the Chromium Next GEM Single Cell 5' kit (10X Genomics) according to the manufacturer's protocol. Purified libraries sequenced together on one S4 lane of a NovaSeq sequencing instrument with the run configuration $150 \times 10 \times 10 \times 150$.

## scRNAseq analysis

scRNAseq FASTQ files were processed with Cell Ranger v. 7.1.0 (10x Genomics) using default settings for 5' RNA gene expression analysis. Using Seurat (v. 4.4.0) in R, we selected for high-quality transcriptomes by filtering with the following criteria: 500 to 6000 features (detected genes), 2000 to 40,000 unique modular identifier (UMI) counts, ribosomal gene content between 5 and 50%, and mitochondrial gene content below 5%. Next, we used the scGate package to select

target cells that were Cd8a and Cd8b1 positive, resulting in 1372 and 6775 high-quality $Lrp10^{+/+}$ and $Lrp10^{-/-}$ TIL transcriptomes, respectively. 103 $Lrp10^{+/+}$ and 508 $Lrp10^{-/-}$ doublets were predicted and removed using DoubletFinder. Next, we pooled the data (7536 transcriptomes) from the filtered feature matrices and added the corresponding scTCRseq clonotype data into the metadata of the Seurat objects using the filtered contig annotations and clonotype csv files. After the removal of a cluster of contaminating cells expressing NK markers, we performed standard normalization and scaling of the remaining 7333 transcriptomes and identification of 1000 highly variable genes (HVGs) using the vst method in Seurat. Dimensionality reduction on the HVGs was achieved using principal component analysis (PCA) and UMAP on the first 15 principal components. Unsupervised clustering using the Louvain algorithm was implemented using the FindNeighbors and FindClusters functions in Seurat with resolution 0.2. Differentially expressed genes (DEGs) between clusters were identified using FindAllMarkers with the parameters min.pct = 0.25, logfc.threshold = 0.25 and excluding mitochondrial, ribosomal, heat shock, and cell cycle genes. Using the dplyr package and DoHeatmap, we generated a heatmap of the top 15 genes by average fold change in each cluster. Distributions of individual genes signatures and clonotypes in the UMAP were visualized using the FeaturePlot function. Stacked bar graphs were generated using the dittoBarPlot function in the dittoSeq package (https://rdrr.io/bioc/dittoSeq/). The subset of 1673 transcriptomes with clonotypes greater than 2 was extracted and clustered using the first 30 principal components at 0.25 resolution. FindMarkers was used to find the DEGs between the $Lrp10^{+/+}$ and $Lrp10^{-/-}$ cells and FindAllMarkers was used to identify DEGs between clusters. We used EnhancedVolcano (https://github.com/kevinblighe/EnhancedVolcano) with fold change cutoff of 0.5 and $P$ value cutoff of 0.05 to visualize the DEGs between the $Lrp10^{+/+}$ and $Lrp10^{-/-}$ cells.

## scTCRseq analysis

scTCRseq FASTQ files were processed with Cell Ranger v. 7.1.0 (10x Genomics) using default settings with the Single Cell V(D)J-T (Alpha Beta) library. We analyzed clonal proportions using the repClonality function with the "rare" method in the Immunarch 1.0.0 R package (ImmunoMind) to segregate the data according to default bin settings (clonotype counts = 1, 2–3, 4–10, 11–30, 31–100, 101-MAX).

## Generation of anti-Lrp10 monoclonal antibodies

To generate anti-Lrp10 monoclonal antibodies, $Lrp10^{-/-}$ mice were immunized with recombinant murine Lrp10-ECD in alum adjuvant followed by two protein-only boosts every two weeks. Anti-Lrp10 serum titers were monitored by ELISA. After the final boost, the spleen of the mouse with the highest antibody titers was dissociated and fused with Sp2/0-Ag14 myeloma cells (ATCC) using the ClonaCell-HY Hybridoma Kit (StemCell Technologies) according to the manufacturer's protocol. The fusion product was plated in semi-solid selection media and clonal outgrowths were harvested into single wells of 96-well plates after 10–14 days. Single clones were allowed to grow for another 14 days in suspension culture after which the supernatant was tested for reactivity against Lrp10-ECD using ELISA. Wells that scored positive in the ELISA assay were then evaluated by flow cytometry against cell lines that overexpressed full-length Lrp10. Wells that scored positive in the

ELISA and FACS assays were propagated and rescreened. Three different hybridomas that produced anti-Lrp10 antibodies were generated from a single mouse of which only one (clone 6H5) could detect the protein by Western blot. The monoclonality of this hybridoma was verified by sequencing. Antibody was purified using a Hi-Trap Protein G column (Cytiva). Notably, although all three antibodies were screened for FACS reactivity on intact cells, they could only detect cell surface Lrp10 when it was overexpressed. None of them could detect cell surface Lrp10 on primary CD8 T cells.

## Study design and statistical analysis

No statistical methods were used to pre-determine sample size. The investigators were not blinded to experimental groups while collecting data. Mice were randomly allocated for experimentation and analysis if they met the genotype, age, and gender requirements of the study. For TIL analysis, a subset of tumor-bearing mice from each group were randomly selected for analysis without specific inclusion or exclusion criteria. The normal distribution of data was determined by the Shapiro–Wilk normality test. For normally distributed data, the statistical significance of differences between experimental groups was determined by paired or unpaired $t$ tests as indicated. For non-normally distributed data, a nonparametric test was used as indicated. Multiple comparisons were analyzed with ANOVA. Statistical analyses were performed using GraphPad Prism software. Differences with $P$ values < 0.05 were considered significant. Differences with $P$ values ≥0.05 were considered not significant (ns). Exact $P$ values are indicated in the figures.

# Data availability

The raw and processed scRNA-seq and scTCRseq files for this analysis are available at the Gene Expression Omnibus (GEO) under the accession number GSE264241.

The source data of this paper are collected in the following database record: biostudies:S-SCDT-10_1038-S44319-024-00191-w.

# Peer review information

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

## Acknowledgements

The authors thank Emre Turer, Anne Satterthwaite, and Tuoqi Wu for critical reading of this manuscript. The authors thank Sook Kyung Chang and Elsam Elghonaimy for helpful discussions. The authors also thank Kennith Stedham for his help with managing the mouse colony. We acknowledge Caitlin Eaton and Vanessa Schmid from the UTSW McDermott Next Generation Sequencing Core for library preparation and sequencing during the single-cell experiments. Analysis of the single-cell RNA sequencing data was aided by computational resources from the BioHPC supercomputing facility located in the Lyda Hill Department of Bioinformatics, UT Southwestern Medical Center. Initial portions of this work were funded by a sponsored research agreement with ImmunoDesigners, Inc. This work was funded by NIH R01-AI125581 (BB) and NIH R01-AI167920 (EN-G).

## Author contributions

**Jamie Russell**: Data curation; Investigation; Methodology. **Luming Chen**: Data curation; Formal analysis; Validation; Investigation; Visualization; Methodology; Writing—review and editing. **AiJie Liu**: Investigation; Methodology. **Jianhui Wang**: Investigation; Methodology. **Subarna Ghosh**: Investigation. **Xue Zhong**: Investigation; Methodology. **Hexin Shi**: Investigation. **Bruce Beutler**: Conceptualization; Resources; Funding acquisition; Methodology. **Evan Nair-Gill**: Conceptualization; Resources; Data curation; Formal analysis; Supervision; Funding acquisition; Validation; Investigation; Visualization; Methodology; Writing—original draft; Project administration; Writing—review and editing.

Source data underlying figure panels in this paper may have individual authorship assigned. Where available, figure panel/source data authorship is listed in the following database record: biostudies:S-SCDT-10_1038-S44319-024-00191-w.

## Disclosure and competing interests statement

XZ, BB, and EN-G have licensed intellectual property and have received royalties related to Lrp10.

# Expanded View Figures

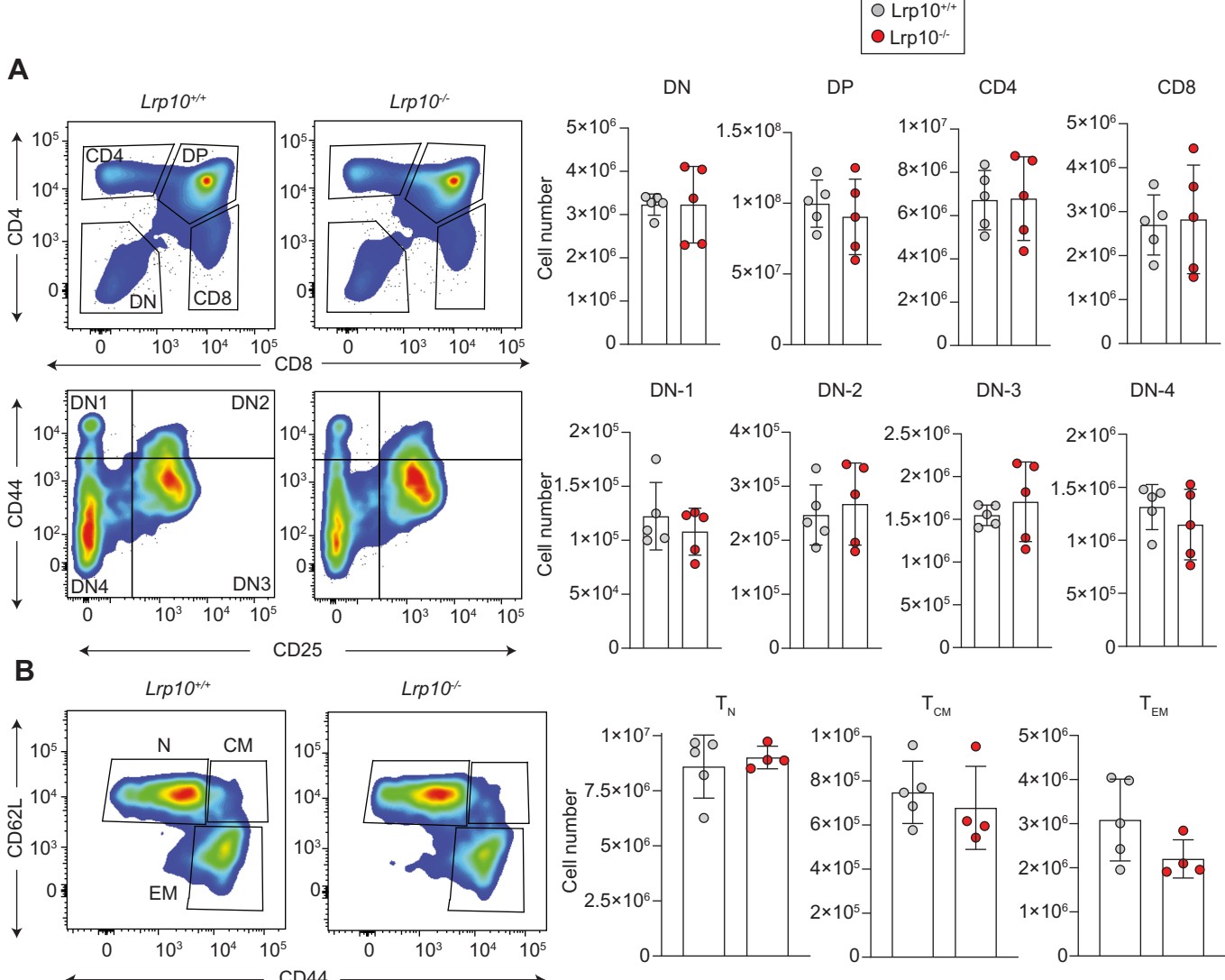

**Figure EV1. Numbers of thymocyte and peripheral CD4 T cell subpopulations in *Lrp10⁻/⁻* mice.**

(A) Representative FACS plots and numbers of thymic T-cell subpopulations. (B) Representative FACS plots and numbers of splenic CD4 subpopulations. Data information: In bar graphs, symbols represent individual mice (biological replicates), horizontal bars indicate the mean, and error bars show SD. No statistical testing was performed.

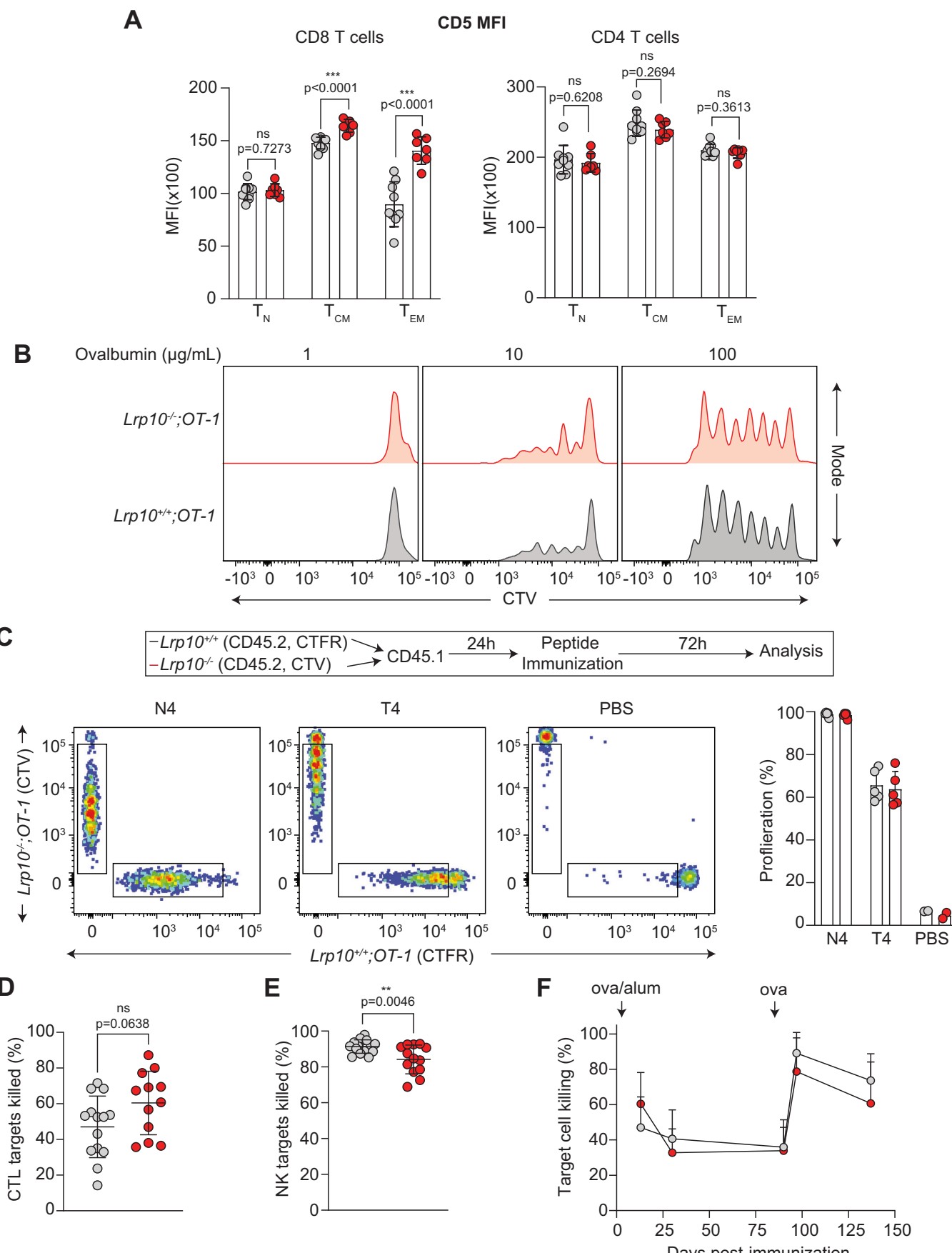

**Figure EV2.   TCR sensitivity and cytotoxic activity in *Lrp10⁻/⁻* mice.**

(A) Cell surface CD5 expression in splenic CD8 and CD4 T-cell subsets. (B) Proliferation of *Lrp10⁺/⁺;OT-1* or *Lrp10⁻/⁻;OT-1* cells 72 h after co-culture with *Lrp10⁺/⁺* dendritic cells pulsed with the indicated concentrations of ovalbumin. (C) $10^6$ *Lrp10⁺/⁺;OT-1* and *Lrp10⁻/⁻;OT-1* cells were labeled with CTFR and CTV, respectively, and injected into unirradiated recipients. 24 h later mice were immunized with SIINFEKL (N4), SIITFEKL (T4), or PBS. Cell proliferation was measured 72 h after immunization based on dye dilution. (D) In vivo CTL cytotoxicity in *Lrp10⁺/⁺* and *Lrp10⁻/⁻* mice 12 days after immunization with ovalbumin and aluminum adjuvant (ova/alum). $N = 14$ *Lrp10⁺/⁺* and $n = 12$ *Lrp10⁻/⁻* mice. (E) In vivo NK cytotoxicity assay in *Lrp10⁺/⁺* and *Lrp10⁻/⁻* mice injected with labeled MHC-I-deficient target cells. $N = 15$ *Lrp10⁺/⁺* and $n = 13$ *Lrp10⁻/⁻* mice. (F) Serial in vivo cytotoxicity assays in the mice from (E) after immunization with ova/alum. Mice were given a boost of ova protein alone on day 90. $N = 14$ *Lrp10⁺/⁺* and $n = 12$ *Lrp10⁻/⁻* mice at each timepoint. Data information: In bar graphs, symbols represent individual mice (biological replicates), horizontal bars indicate mean values, and error bars show SD. In (F), symbols indicate the mean values and error bars show SD. Data from (B) is representative of two separate in vitro stimulation experiments. Data from (D, E) were replicated twice on separate cohorts of at least 10 mice. Data from (C, F) are from one immunization experiment each with the indicated *n*. *P* values were calculated by two-tailed unpaired *t* tests (A, D, E). No statistical testing was performed in (C, F). Significant *P* values were flagged as follows: *$P < 0.05$, **$P < 0.01$, ***$P < 0.001$. *P* values $> 0.05$ were considered to be not significant (ns).

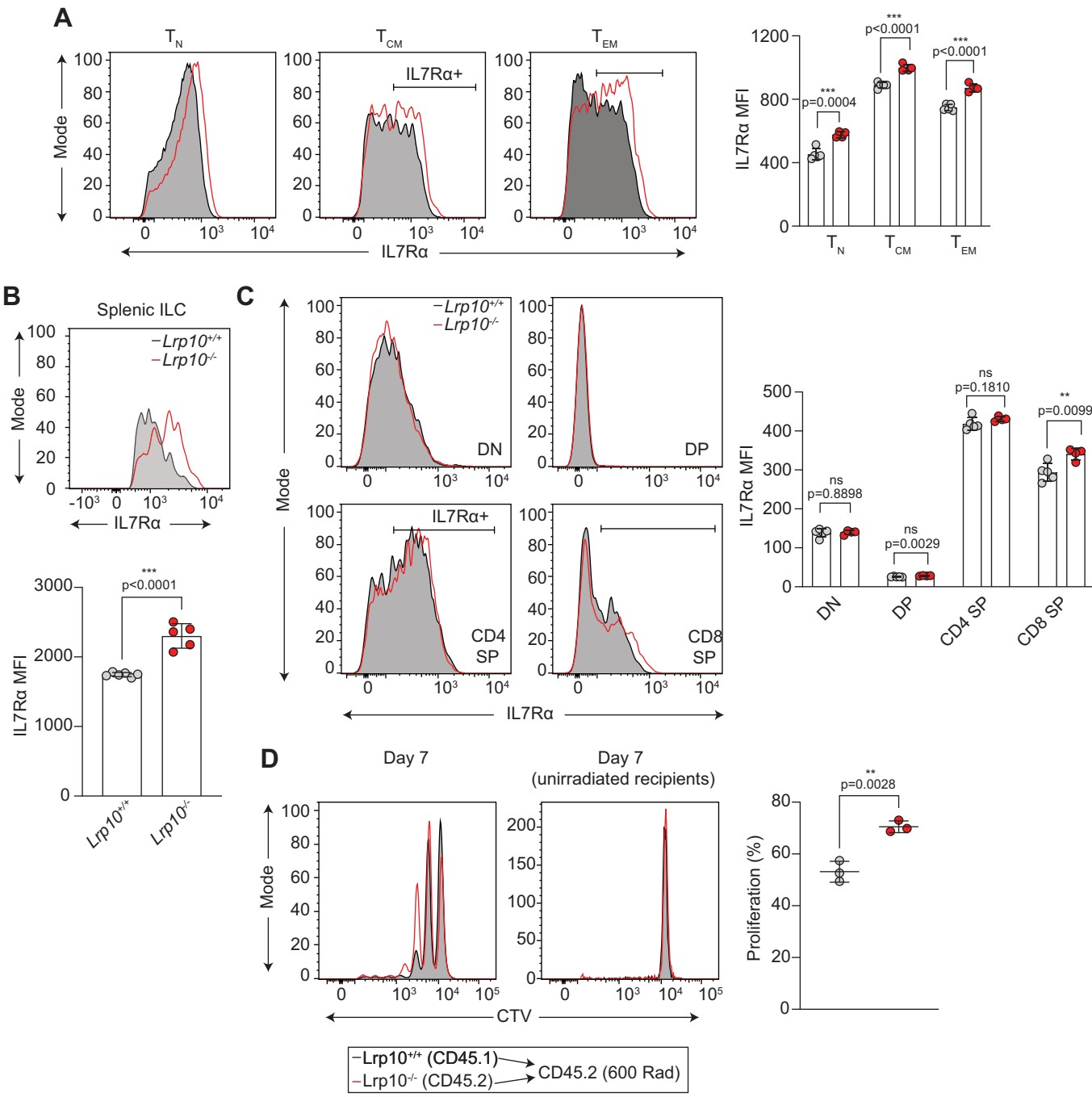

**Figure EV3. IL7R expression on CD4 T cell, ILC, and thymocyte subpopulations.**

(A) Cell surface IL7R expression in CD4 subpopulations. (B) Cell surface IL7R expression on Lin-CD11b-IL7R + NK1.1 + NKp46+ splenic ILCs. Lin: CD3, B220, CD11c. (C) Cell surface IL7R expression on thymic T-cell subpopulations. (D) Representative FACS plots of homeostatic expansion of *Lrp10*+/+ and *Lrp10*-/- CD4 T cells labeled with CTV and injected in sub-lethally irradiated and unirradiated recipients. The bar graph shows proliferation of CD4 T cells transplanted into irradiated recipients. Proliferation was determined based on the fraction of cells that underwent at least one cell division. Data information: In bar graphs, symbols represent individual mice (biological replicates), horizontal bars indicate the mean, and error bars show SD. Data from (D) are from one CD4 adoptive transfer experiment with three irradiated recipients and one unirradiated recipient. *P* values were calculated by two-tailed unpaired t tests. Significant p values were flagged as follows: *$P < 0.05$, **$P < 0.01$, ***$P < 0.001$. *P* values $> 0.05$ were considered to be not significant (ns).

**A**

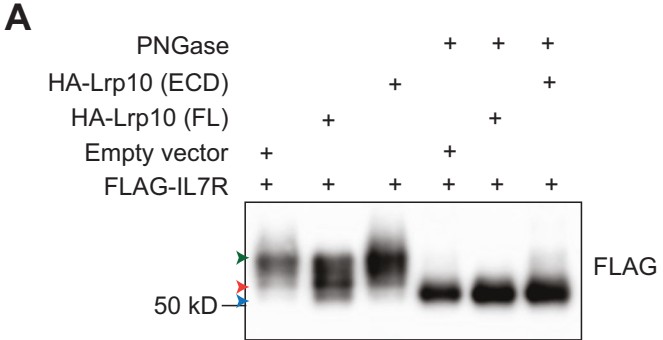

**B**

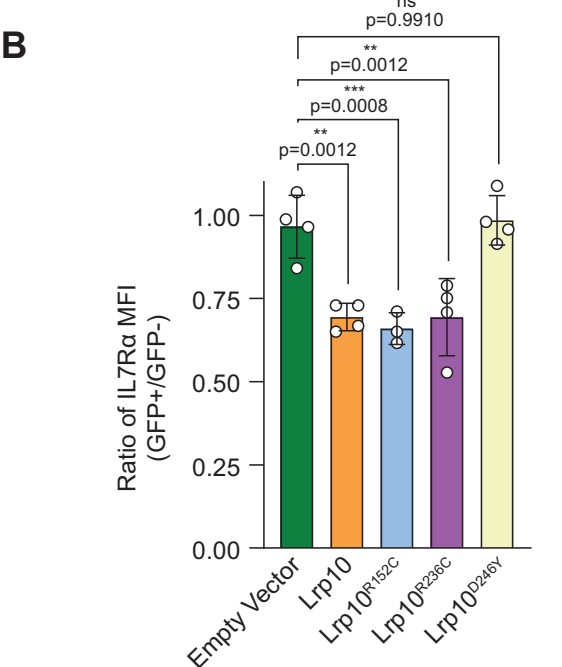

**Figure EV4.** **Effect of Lrp10 variants on IL7R glycosylation and cell surface expression.**

(A) IP of IL7Rα-FLAG from transfected HEK 293 T cells under denaturing conditions in the context of empty vector, Lrp10-HA (FL), or Lrp10-HA (ECD) followed by de-glycosylation with PNGase-F. (B) Normalized cell surface IL7R expression on activated *Lrp10$^{-/-}$* CD8 T cells infected with MSCV retroviruses encoding GFP-only, Lrp10$^{WT}$-IRES-GFP, Lrp10$^{R132C}$-IRES-GFP, Lrp10$^{R235C}$-IRES-GFP, or Lrp10$^{D246Y}$-IRES-GFP (*chowmein* allele). IL7R levels on GFP+ cells in each sample were normalized to levels on the GFP- population. Data information: In (B), the symbols show the results of three or four separate retroviral transductions (biological replicates), the horizontal bars indicate mean values, and error bars show SD. *P* values were calculated by one-way ANOVA with Dunnett's multiple comparisons test. Significant *P* values were flagged as follows: *$P < 0.05$, **$P < 0.01$, ***$P < 0.001$. *P* values > 0.05 were considered to be not significant (ns).

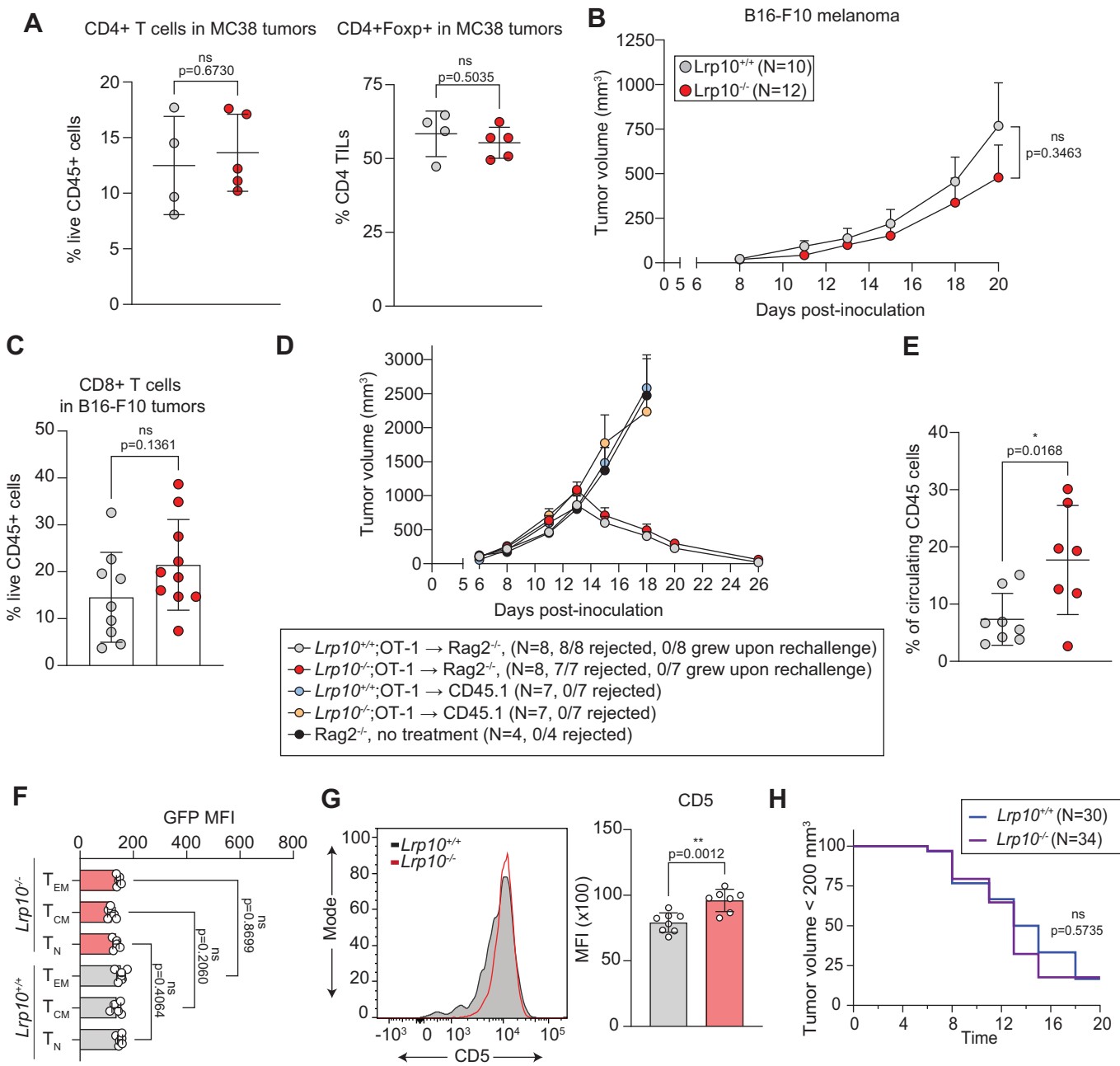

**Figure EV5.  Effect of *Lrp10* deletion on tumor growth in poorly immunogenic tumors and during adoptive transfer.**

(A) Frequency of total CD4 T cells and CD4+Foxp3+ Tregs in total CD4 that were infiltrating D18 MC38 tumors on *Lrp10⁺/⁺* and *Lrp10⁻/⁻* mice. (B) Tumor growth of B16F10 melanoma cells injected subcutaneously into *Lrp10⁺/⁺* and *Lrp10⁻* mice. (C) Frequency of CD8 T cells infiltrating B16F10 tumors on D20. (D) Adoptive transfer of *Lrp10⁺/⁺;OT-1* or *Lrp10⁻/⁻;OT-1* cells into the indicated recipients on D6 after inoculation with B16-ova cells. The number of *Rag2⁻/⁻* recipients that rejected the primary tumor and were resistant to tumor rechallenge 30 days later is indicated. (E) The frequency of *Lrp10⁺/⁺;OT-1* or *Lrp10⁻/⁻;OT-1* cells in the peripheral blood of *Rag2⁻/⁻* recipients 30 days after primary tumor rejection. (F) GFP MFI in splenic CD8 T-cell subsets from $n = 6$ naive *Lrp10⁺/⁺;Nur77^GFP* and $n = 5$ *Lrp10⁻/⁻;Nur77^GFP* mice. (G) Representative FACS plot and CD5 MFI in T_CM phenotype cells from MC38 tumors on D18. (H) Frequency of B16F10 melanoma tumor progression in *Lrp10⁺/⁺* and *Lrp10⁻/⁻* mice treated with 10 mg/kg anti-PD1 across three separate cohorts with the indicated number n per genotype. Mice were said to have progressed if tumor volume exceeded 200 mm³. Data information: In bar graphs, symbols represent individual mice (biological replicates), horizontal bars indicate the mean, and error bars show SD. In (B, D), symbols represent the mean value and error bars show SEM. Results shown in (A, B, D, E, G) were replicated twice (A, D, E, G) or three times (B) in separate cohorts of at least three (A) or five mice (B, D, E, G) per genotype. Results shown in (C) are combined from two separate experiments. P values calculated with two-tailed unpaired t tests in (A, B, C, E, G), one-way ANOVA with Tukey's test (F), and log-rank (Mantel–Cox) test (H). Significant P values were flagged as follows: *P < 0.05, **P < 0.01, ***P < 0.001. P values > 0.05 were considered to be not significant (ns).

