## [Peer Review File · EMBO Reports]

Lrp10 suppresses IL7R limiting CD8 T cell homeostatic expansion and anti-tumor immunity.

Jamie Russell, Luming Chen, Aijie Liu, Jianhui Wang, Subarna Ghosh, Xue Zhong, Hexin Shi, Bruce Beutler, and Evan

Nair-Gill *Corresponding author(s): Evan Nair-Gill (Evan.Nair-Gill@UTSouthwestern.edu)*

Review Timeline:

Submission Date:	14th Dec 23
Editorial Decision:	26th Jan 24
Revision Received:	30th Apr 24
Editorial Decision:	23rd May 24
Revision Received:	7th Jun 24
Accepted:	17th Jun 24

Editor: Achim Breiling

Transaction Report:

Dear Dr. Nair-Gill,

Thank you for the transfer of your research manuscript to EMBO reports. I have now received the reports from the three referees that were asked to evaluate your study, which can be found at the end of this email.

As you will see, the referees think that the findings are of high interest. However, they have several comments, concerns, and suggestions, indicating that a major revision of the manuscript is necessary to allow publication of the study in EMBO reports. As the reports are below, and all the referee concerns need to be addressed, I will not detail them here.

Given the constructive referee comments, I would like to invite you to revise your manuscript with the understanding that all referee concerns must be addressed in the revised manuscript or in a detailed point-by-point response. Acceptance of your manuscript will depend on a positive outcome of a second round of review. It is EMBO reports policy to allow a single round of revision only and acceptance of the manuscript will therefore depend on the completeness of your responses included in the next, final version of the manuscript.

- 1) a .docx formatted version of the final manuscript text (including legends for main figures, EV figures and tables), but without the figures included. Figure legends should be compiled at the end of the manuscript text.
- 2) individual production quality figure files as .eps, .tif, .jpg (one file per figure), of main figures (up to 8) and EV figures (up to 5). Please upload these as separate, individual files upon re-submission.

- 4) a complete author checklist, which you can download from our author guidelines (<https://www.embopress.org/page/journal/14693178/authorguide>). Please insert page numbers in the checklist to indicate where the requested information can be found in the manuscript. The completed author checklist will also be part of the RPF.

- 5) that primary datasets produced in this study (e.g. RNA-seq, ChIP-seq, structural and array data) are deposited in an

appropriate public database. If no primary datasets have been deposited, please also state this in a dedicated section (e.g. 'No primary datasets have been generated and deposited'), see below.

The accession numbers and database should be listed in a formal "Data Availability" section (placed after Materials & Methods) that follows the model below. This is now mandatory (like the COI statement). Please note that the Data Availability Section is restricted to new primary data that are part of this study. This section is mandatory. As indicated above, if no primary datasets have been deposited, please state this in this section

Data availability

8) Regarding data quantification and statistics, please make sure that the number "n" for how many independent experiments were performed, their nature (biological versus technical replicates), the bars and error bars (e.g. SEM, SD) and the test used to calculate p-values is indicated in the respective figure legends (also for potential EV figures and all those in the final Appendix). Please also check that all the p-values are explained in the legend, and that these fit to those shown in the figure. Please provide statistical testing where applicable. Please avoid the phrase 'independent experiment', but clearly state if these were biological or technical replicates. Please also indicate (e.g. with n.s.) if testing was performed, but the differences are not significant. In case n=2, please show the data as separate datapoints without error bars and statistics. See also: <http://www.embopress.org/page/journal/14693178/authorguide#statisticalanalysis>

9) Please add scale bars of similar style and thickness to microscopic images, using clearly visible black or white bars (depending on the background). Please place these in the lower right corner of the images themselves. Please do not write on or near the bars in the image but define the size in the respective figure legend.

10) Please also note our reference format:

12) We now use CRedit to specify the contributions of each author in the journal submission system. CRedit replaces the author contribution section. Please use the free text box to provide more detailed descriptions and do not provide your final manuscript text file with an author contributions section. See also our guide to authors: <https://www.embopress.org/page/journal/14693178/authorguide#authorshipguidelines>

13) We would encourage you to use 'Structured Methods', our new Materials and Methods format. According to this format, the

Materials and Methods section should include a Reagents and Tools Table (listing key reagents, experimental models, software and relevant equipment and including their sources and relevant identifiers), uploaded as separate file, followed by a Methods and Protocols section in which we encourage the authors to describe their methods using a step-by-step protocol format with bullet points, to facilitate the adoption of the methodologies across labs. More information on how to adhere to this format as well as downloadable templates (.doc or .xls) for the Reagents and Tools Table can be found in our author guidelines (section 'Structured Methods'):

14) Please add up to 5 keywords to the manuscript text file order the manuscript sections like this, using these names: Title page - Abstract - Keywords - Introduction - Results - Discussion - Materials and Methods - Data availability section - Acknowledgements - Disclosure and Competing Interests Statement - References - Figure legends - Expanded View Figure legends

Finally, please note that all corresponding authors are required to supply an ORCID ID for their name upon submission of a revised manuscript. Please find instructions on how to link the ORCID ID to the account in our manuscript tracking system in our Author guidelines: <http://www.embopress.org/page/journal/14693178/authorguide#authorshipguidelines>

I look forward to seeing a revised version of your manuscript when it is ready. Please let me know if you have questions or comments regarding the revision.

Yours sincerely,

Referee #1:

In the manuscript by Russell and colleagues, the authors investigated a role for Lrp10 in T cells. The major role identified in this study is a role for Lrp10 in suppressing IL7R expression post-translationally. Lrp10 deficiency results in accumulation of naive and central memory phenotype cells and an augmented anti-tumor response, attributed to a shift in the transcriptional phenotypes of tumor-infiltrating T cells away from a terminally differentiated program and toward a more memory-like phenotype. This work is well-designed and executed and describes a novel role for Lrp10 in T cell immunity. In my opinion this work could be acceptable for publication if the following critiques are addressed.

1. The authors imply that the inhibitory effect of Lrp10 on IL7R expression is mediated through the Lrp10 extracellular domain (ECD). For Figure 3C, the proper control should be to co-IP with the Lrp10 mutant lacking ECD) to show that co-precipitation is abolished. To solidify this point, the authors could alternatively show that an IL-7R lacking its extracellular domain also does not co-precipitate with the ECD of Lrp10.
2. It would be important for the authors to expand their discussion to include more information about the intracellular domain of Lrp10 and any amino acid sequence motifs that could promote the post-translational modification of IL7R.
3. The singlet memory-like T cell clones in Lrp10^{-/-} tumors are intriguing. If these T cells are cross-reactive with self-and tumor-associated antigens, do these cells express elevated levels of the receptor CD5? High CD5 expression in T cells is associated with increased self-reactivity.
4. Related to the point above, it is interesting that the intensity of Nur77-GFP expression is relatively increased in Lrp10^{-/-} tumor-resident T cells relative to WT. This result suggests that the T cells have experienced TCR stimulation in the tumor, as mentioned. To aid in interpretation of this result, it would help to see baseline Nur77-GFP expression in naive T cells from untreated mice. Are there any differences in Nur77-GFP MFI between naive WT and Lrp10^{-/-} T cells in the secondary lymphoid organs?
5. It doesn't appear that there are legends for Tables 1 and 2.

Referee #2:

In this study, Russel et al analyze the phenotype of a KO mouse strain obtained by ENU mutagenesis. This KO targets the gene

Lrp10 that had no known function in the immune system. The authors found an increased number of CD8 T cells and a decrease in NK cells. They established that Lrp10 is involved in the post-translational regulation of the IL7R, involved in the maintenance of Naïve and Memory T cells, explaining the increase in CD8 T cell number. They nicely established an increased sensitivity to IL-7 for Lrp10ko CD8 T cells and then focus on the consequences of Lrp10 KO during anti-tumor response. Increased level of IL7R led to a less exhausted phenotype of tumor specific CD8 T cells and increased efficiency of PD-1 blockade using MC38 tumor cell line. The paper is clear, thorough and explores nicely the role of a protein that had not been studied in the immune system before. This is a really interesting study. I have a few comments that should be addressed before publication.

The authors mention the lack of autoimmunity in this mouse model. They should illustrate this and describe/show what they looked at to conclude to a lack of autoimmunity.

The initial description of the effect of Lrp10 inactivation on the immune system cell types showed CD8, CD4, NK1.1, B220 and CD11b cell numbers. Is there any effect on dendritic cells? One important phenotype is a decrease in NK1.1 cells. The authors should also look at other members of the ILC family. They are characterized by high level of CD127 and could be affected in the KO.

Related to this question, is the level of CD127 different in WT and KO NK cells?

One surprising result is the lack of effect in CD4 T cells. Could the author discuss this point? Why Lrp10 is more active in CD8 than in CD4 T cells?

The author showed a difference in response to anti-PD1 treatment between Lrp10 WT and KO mice. The initial response was already different without PD-1 blockade. Has PD-1 blocking a different outcome using B16 melanoma where the KO had no initial effect on tumor growth?

As a minor comment, in line 131 the authors mention IL7 instead of IL7R.

Referee #3:

In this manuscript Russell et al. describe the role of Lrp10 in the regulation of CD8+ T cell homeostatic expansion and anti-tumor immunity. They identified Lrp10 as a new determinant of immune cell homeostasis via a forward genetic screening. Then they generated a total body Lrp10^{-/-} mouse model. This was used together with classic tumor immunology experiments and single cell transcriptomic/single cell TCR sequencing analysis to define the heterogeneity of tumor infiltrating Lrp10 Wt and Lrp10 KO CD8 T cells. Moreover, although in a different (and less physiologically relevant) cellular model, the authors mechanistically study the interaction between Lrp10 and IL7R.

Overall, the authors claim that Lrp10 works as a negative regulator of IL7R and that genetic deletion of Lrp10 favors CD8 T cells tumor infiltration, cytokine production and eventually tumor clearance together with reduced T cell exhaustion over time in an IL7R-dependent manner in immunogenic hot tumors. One last observation suggests Lrp10 reduced expression might synergize with anti-PD1 immunotherapy.

The work submitted by Russell et al. delves into a very interesting field presenting new data on a protein previously unrelated with immune function. In general, the conclusions are supported by the results and are very elegantly presented. Moreover, the data present a clear translational perspective although future work is needed to identify the binding partners of Lrp10 and add more mechanistic insights on how Lrp10 regulates T cell function to progress in more clinical settings.

The authors also highlight themselves a possible critical limitation of study when they underline that their model is a full body KO and a possible role for other immune cells cannot be ruled out.

Some suggestions from this reviewer are the following:

- Pathogenic variants of Lrp10 have been associated with Parkinson and ALS. While I understand that the generation and characterization of new point mutation mouse models is of course beyond the scope of this project, it would be interesting to check in a reconstitution experiment (similar to the ones already performed by the authors) the effects of these pathogenic variants on IL7R binding and expression.
- Total body Lrp10 KO in mice is overall protective against tumor because of the CD8 mechanism described by the authors. Are there any publicly available data to generate Kaplan-Meier plot assessing correlation of Lrp10 expression in patients with hot vs cold tumors and their survival?
- The authors are well aware of a possible contribution of other Lrp10 KO cells to the higher tumor infiltrating and clearance potential of Lrp10 CD8 KO cells in the total body KO model. Nevertheless, an experiment that would help solve this question is possible and well in the capabilities of the authors in terms of mouse strains and tumor models available and would consist in adoptively transfer Lrp10^{+/+} and Lrp10^{-/-} CD8 OT-I cells in syngeneic mice previously inoculated with EL4-OVA, B16-OVA and MC38-OVA tumors and check tumor progression or remission.

Minor:

- A reference in the main text to Fig S2B is missing
- Line 216, S4A refers to % of CD4 cells in the CD45+ subset and this is not precisely mentioned in the text that instead describes S4B and the FoxP3+ in the CD4 subset.

Dear Editor,

Please find our revised manuscript attached. We thank the reviewers for their insightful comments that have helped us to improve this manuscript. From their suggestions we have added significant new data that increases our understanding of the immune phenotype and mechanism of Lrp10. We have also substantially expanded the discussion section to address important considerations of Lrp10 in autoimmunity, anti-tumor responses, and the regulation of IL7R expression. Our detailed point-by-point response to their critiques is below.

Referee #1:

In the manuscript by Russell and colleagues, the authors investigated a role for Lrp10 in T cells. The major role identified in this study is a role for Lrp10 in suppressing IL7R expression post-translationally. Lrp10 deficiency results in accumulation of naive and central memory phenotype cells and an augmented anti-tumor response, attributed to a shift in the transcriptional phenotypes of tumor-infiltrating T cells away from a terminally differentiated program and toward a more memory-like phenotype. This work is well-designed and executed and describes a novel role for Lrp10 in T cell immunity. In my opinion this work could be acceptable for publication if the following critiques are addressed.

1. The authors imply that the inhibitory effect of Lrp10 on IL7R expression is mediated through the Lrp10 extracellular domain (ECD). For Figure 3C, the proper control should be to co-IP with the Lrp10 mutant lacking ECD) to show that co-precipitation is abolished. To solidify this point, the authors could alternatively show that an IL-7R lacking its extracellular domain also does not co-precipitate with the ECD of Lrp10.

Response: We thank the reviewer for this insightful and helpful comment. Based on the suggested domain analysis of Lrp10 and IL7R we have updated our model. Our initial data showed that Lrp10 suppressed IL7R expression in reconstituted Lrp10 KO CD8 T cells and in the heterologous 293T expression system. We have expanded this dataset to show that Lrp10 suppresses IL7R expression in an Lrp10 dose-dependent manner (Fig. 3C). We then co-expressed Lrp10-full length (FL), Lrp10-ECD, Lrp10-ICD/TM (expressing the ICD and transmembrane domain, TM), and Lrp10 ICD (expressing ICD only) (Fig. 3D). Lrp10-ECD did not suppress IL7R. Surprisingly, Lrp10-ICD/TM potently suppressed IL7R. This effect was partially reversed when IL7R was co-transfected with Lrp10 ICD (additional deletion of the TM domain). These data indicate that the suppressive effect of Lrp10 on IL7R is mediated by the ICD and potentiated by the presence of the TM region.

We went on to test binding between Lrp10 and IL7R (Fig. 3E). We found that Lrp10-FL, Lrp10-ECD, Lrp10 ICD/TM, and Lrp10 ICD could all bind to IL7R to some extent, although the Lrp10 ICD qualitatively had a higher level of binding compared to the full-length protein or the ECD. Binding to all deletion mutants of Lrp10 was strongly attenuated by deletion of the IL7R ECD. It is difficult to reconcile both the ECD and the ICD of Lrp10 (which exist on either side of the plasma membrane) binding directly to the IL7R-ECD. Therefore, we suspect that Lrp10 is part of a membrane-bound protein complex that is recruited to IL7R to suppress its expression. The identity and nature of such a complex remains to be discovered. Based on available protein interactome data (Bioplex 3.0), Lrp10 may interact with several immunologically important receptors. Therefore, while the relationship between Lrp10 and IL7R is important for the T cell phenotypes presented in this paper, it is possible that Lrp10 may affect the expression and function of diverse receptors. It will be interesting to test in the future if these effects are mediated by the Lrp10 ICD.

2. It would be important for the authors to expand their discussion to include more information about the intracellular domain of Lrp10 and any amino acid sequence motifs that could promote the post-translational modification of IL7R.

Response: While Lrp10 suppresses IL7R through its ICD, the exact mechanism remains unclear. We note that Lrp10 prevents expression of the fully glycosylated mature form of IL7R (Fig. EVE4A). This may happen through two possible mechanisms: (1) it interferes with IL7R maturation in the Golgi (for example by blocking access to protein glycosyl-transferases), or (2) it triggers the destruction of mature IL7R. Interestingly, the Lrp10 ICD is proline rich and contains a PPxY motif. These motifs are known to bind to WW domains found in HECT-family NEDD4-like E3 ubiquitin ligases (PMID: 26245901). PPxY

binding to WW domains disrupts the auto-inhibited form of NEDD4-like E3 ligases and augments ubiquitin-ligase activity (PMID: 26245901). Notably, the Lrp10 relative LDLRAD3 was previously shown to activate the NEDD4-like E3 ligase Itch through its two PPxY motifs (PMID: 26854353). Additionally, the Lrp10 interactome from Bioplex 3.0 suggests that Lrp10 can interact with Nedd4-like family members (WWP2). Whether the PPxY motif of Lrp10 activates NEDD4-like E3 ligases to target IL7R for ubiquitination is a topic of active investigation in our lab.

Previous studies have shown that Lrp10 traffics between the trans-Golgi, plasma membrane, and endosomes. It is unclear where in its membrane trafficking itinerary Lrp10 interferes with IL7R. Two DXXL motifs (AP1/AP2 and GGA binding motifs) in the Lrp10 ICD mediate receptor transport from plasma membrane to the endosomes and retrograde transport from the endosome to the trans-Golgi (PMID: 18627575). Mutation of these motifs were previously shown to increase retention of Lrp10 at the cell surface and within early endosomes (PMID: 22734645). It will be interesting to see how mutation of these DXXL motifs change IL7R expression. If mutating the Lrp10 DXXL motifs increases IL7R expression, it could suggest that Lrp10 promotes IL7R removal from the cell surface and its delivery to the endo/lysosomal system.

These points were added to the Discussion section lines 453-470.

3. The singlet memory-like T cell clones in Lrp10^{-/-} tumors are intriguing. If these T cells are cross-reactive with self-and tumor-associated antigens, do these cells express elevated levels of the receptor CD5? High CD5 expression in T cells is associated with increased self-reactivity.

Response: We appreciate this suggestion by the reviewer. We analyzed CD5 expression within the Tcm phenotype population within MC38 tumors (CD62L⁺CD44⁺). In agreement with these cells having higher levels of self-reactivity, they showed elevated CD5 cell surface expression levels (Fig. EV5G). We also looked at the spleen of naïve mice and found that CD8 Tcm and Tem cells, but not naïve cells, displayed higher CD5 levels in Lrp10 KO mice (Fig. EV2A). CD4 cells did not show any differences in CD5 levels. These data demonstrate that Lrp10 KO mice maintain a pool of CD8 memory T cells expressing higher levels of CD5 thus indicating increased self-reactivity. We suspect that maintenance of these populations is driven by enhanced IL7R signaling.

4. Related to the point above, it is interesting that the intensity of Nur77-GFP expression is relatively increased in Lrp10^{-/-} tumor-resident T cells relative to WT. This result suggests that the T cells have experienced TCR stimulation in the tumor, as mentioned. To aid in interpretation of this result, it would help to see baseline Nur77-GFP expression in naïve T cells from untreated mice. Are there any differences in Nur77-GFP MFI between naïve WT and Lrp10^{-/-} T cells in the secondary lymphoid organs?

Response: Overall, GFP expression was low in CD8 T cell subsets in unchallenged naïve mice and there was no difference in GFP expression between WT and Lrp10 KO CD8 T cell subsets (Fig. EV5F). Therefore, we conclude that while Lrp10 deletion enriches populations of memory cells that show signs of self-reactivity based on CD5 expression, it does not impart constitutive tonic basal TCR signaling to CD8 T cells.

5. It doesn't appear that there are legends for Tables 1 and 2.

Response: We have added these legends to the supplemental material.

Referee #2:

In this study, Russel et al analyze the phenotype of a KO mouse strain obtained by ENU mutagenesis. This KO targets the gene Lrp10 that had no known function in the immune system. The authors found an increased number of CD8 T cells and a decrease in NK cells. They established that Lrp10 is involved in the post-translational regulation of the IL7R, involved in the maintenance of Naïve and Memory T cells, explaining the increase in CD8 T cell number. They nicely established an increased sensitivity to IL-7 for Lrp10ko CD8 T cells and then focus on the consequences of Lrp10 KO during anti-tumor response. Increased level of IL7R led to a less exhausted phenotype of tumor specific CD8 T cells and increased

efficiency of PD-1 blockade using MC38 tumor cell line. The paper is clear, thorough and explore nicely the role of a protein that had not been studied in the immune system before. This is a really interesting study. I have a few comments that should be addressed before publication.

1. The authors mention the lack of autoimmunity in this mouse model. They should illustrate this and describe/show what they looked at to conclude to a lack of autoimmunity.

Response: We appreciate this question from the reviewer and have revised our statement on autoimmunity in Lrp10 KO. Our initial assumption was based on the fact that we do not see typical signs of spontaneous autoimmunity like dermatitis, arthritis, or reduced survival that are observed in other autoimmune strains (e.g. CTLA-4 KO or Itch KO). However, based on heightened CD5 levels in Lrp10 KO T cells, there is likely an increased frequency of self-reactive T cells. Because of this finding, and that increased IL7R signaling is associated with autoimmunity, we would predict that Lrp10 deletion leads to a lower threshold for autoimmunity. C57Bl/6 mice are generally more resistant to spontaneous autoimmunity compared to other strains like Balb/c or NOD and usually require immunization protocols to induce autoimmunity. In the future, it will be interesting to see if Lrp10 KO mice are more susceptible to induced forms of autoimmunity (e.g., EAE or collagen-induced arthritis). We would predict that they are. Additionally, by targeting Lrp10 on sensitized genetic backgrounds (for example, SKG mice or NOD mice) we may see that Lrp10 deletion worsens the development of spontaneous autoimmunity. Additionally, as discussed in relation to question 4, CD4 T cells are not affected by Lrp10 deletion to the same extent as CD8 T cells. This may be protective against spontaneous autoimmune disease.

We have added these points to the Discussion section in lines 400-427.

2. The initial description of the effect of Lrp10 inactivation on the immune system cell types showed CD8, CD4, NK1.1, B220 and CD11b cell numbers. Is there any effect on dendritic cells? One important phenotype is a decrease in NK1.1 cells. The authors should also look at other members of the ILC family. They are characterized by high level of CD127 and could be affected in the KO.

Response: We took a more careful look at the numbers of innate lymphoid and myeloid cells in the spleens of mutant mice. In the revised dataset we fractionated Ter119-CD3-B220-NK1.1- myeloid cells into CD11c+ dendritic cells, CD11c-CD11b+SSC-low monocytes/macrophages (both Gr1+ and Gr1-), and CD11c-CD11b+Gr1+SSC-high neutrophils according to PMID: 22213571. Overall, we did not detect differences in the numbers of these myeloid populations between Lrp10^{+/+} and Lrp10^{-/-} mice (Fig. 1G).

We also appreciate the question about ILC numbers in Lrp10^{-/-} mice given their known expression of IL7R and also their similarities to the NK lineage, which is diminished upon Lrp10 deletion. ILCs are heterogeneous and their cell surface marker identity is dictated in part by the tissues in which they reside. A phenotypic analysis of ILC subsets in different extra-lymphoid tissues (e.g. intestinal and lung epithelium, or liver) is important to understanding the overall effects of Lrp10 on systemic immunity. However, we felt this type of comprehensive analysis was outside the scope of the current study. We instead focused on splenic ILCs which could be identified based on Lin-(CD3, B220, CD11b, CD11c) NK1.1+NKp46+IL7R+ staining which correlates with an ILC1 identity in mice (PMID: 27328005). We did not detect differences in the number of these ILC cells dependent on Lrp10 genotype. This was in notable contrast to NK cells, which were defined as CD3-B220-CD11c-CD11b+/-NK1.1+NKp46+IL7R- and were strongly diminished in Lrp10 KO mice. It will be important to determine in the future whether tissue-resident ILCs are numerically or functionally different in the absence of Lrp10.

3. Related to this question, is the level of CD127 different in WT and KO NK cells?

Response: Pertaining to question 2, CD127 is not expressed on mature NK cells residing in the spleen and is used as a way to distinguish NK1.1+ NK cells from splenic NK1.1+ ILCs (PMID: 27328005). IL7R is expressed on NK cell during maturation in the thymus. Defining how Lrp10 informs the differentiation trajectory of NK cells is important, and is part of an ongoing study on Lrp10 in NK cells and ILCs. Within the splenic ILC population, we found the level of IL7R was increased with Lrp10 deletion, similar to

what is observed in T cells (Fig. EV3B). We are not yet sure of the ramifications of this finding on ILC development or function.

4. One surprising result is the lack of effect in CD4 T cells. Could the author discuss this point? Why Lrp10 is more active in CD8 than in CD4 T cells?

Response: We did note some effects of Lrp10 deletion on CD4 T cells with respect to IL7R levels and homeostatic expansion (Fig. EV3A and D). However, as pointed out by the reviewer, increased IL7R expression and homeostatic expansion in Lrp10 KO CD4 T cells were less impressive than those observed in Lrp10 KO CD8 T cells. We showed that Lrp10 is induced with CD8 T cell activation (Fig. 1D) and our overall model is that Lrp10 induction limits IL7R expression to suppress the accumulation of Tcm cells. Several lines of evidence have previously shown that the requirements for CD8 T cell activation are quantifiably different than those for CD4 T cells: CD8 T cells have a lower threshold for activation whereas CD4 T cells have an increased requirement for co-stimulatory molecules and specific cytokines (PMID: 12942084). Moreover, most cells in the body express MHC-I thereby providing many opportunities for CD8 stimulation by endogenous or self-antigens. In contrast, MHC-II is restricted to specific subsets of cells. Our data showing that TCR restriction with OT-1 reduces the frequency of CD8 Tcm cells (Fig. 1L) in Lrp10 KO mice suggests that some form of TCR-MHC-I interaction drives the differentiation of CD8 Tcm cells. If CD4 T cells have fewer opportunities for TCR-MHC-II interactions, and are also intrinsically resistant to low levels of TCR stimulation, it may explain why Lrp10 deletion does not have as large an effect on expanding the CD4 compartment. Further work determining the nature of signals that drive Lrp10 expression (e.g. TCR signal strength, co-stimulatory molecules, and cytokine environment) should help clarify the differential effects of Lrp10 deletion on CD8 and CD4 T cells.

We have added these points to the Discussion section in lines 415-427.

5. The author showed a difference in response to anti-PD1 treatment between Lrp10 WT and KO mice. The initial response was already different without PD-1 blockade. Has PD-1 blocking a different outcome using B16 melanoma where the KO had no initial effect on tumor growth?

Response: We performed this experiment on three separate cohorts of mice and overall did not find an increased effect of PD1 blockade in the context of Lrp10 deletion (Fig. EV5H). We showed in the initial dataset that B16 melanomas do not accumulate more CD8 T cells in Lrp10 deficient mice. Therefore, we suspect that an increased number of CD8 T cells need to be present in the tumor in order for PD1 to have an effect.

6. As a minor comment, in line 131 the authors mention IL7 instead of IL7R.

Response: We have fixed the subheading of this section.

Referee #3:

In this manuscript Russell et al. describe the role of Lrp10 in the regulation of CD8+ T cell homeostatic expansion and anti-tumor immunity. They identified Lrp10 as a new determinant of immune cell homeostasis via a forward genetic screening. Then they generated a total body Lrp10^{-/-} mouse model. This was used together with classic tumor immunology experiments and single cell transcriptomic/single cell TCR sequencing analysis to define the heterogeneity of tumor infiltrating Lrp10 Wt and Lrp10 KO CD8 T cells. Moreover, although in a different (and less physiologically relevant) cellular model, the authors mechanistically study the interaction between Lrp10 and IL7R.

Overall, the authors claim that Lrp10 works as a negative regulator of IL7R and that genetic deletion of Lrp10 favors CD8 T cells tumor infiltration, cytokine production and eventually tumor clearance together with reduced T cell exhaustion over time in an IL7R-dependent manner in immunogenic hot tumors. One last observation suggests Lrp10 reduced expression might synergize with anti-PD1 immunotherapy.

The work submitted by Russell et al. delves into a very interesting field presenting new data on a protein

previously unrelated with immune function. In general, the conclusions are supported by the results and are very elegantly presented. Moreover, the data present a clear translational perspective although future work is needed to identify the binding partners of Lrp10 and add more mechanistic insights on how Lrp10 regulates T cell function to progress in more clinical settings.

The authors also highlight themselves a possible critical limitation of study when they underline that their model is a full body KO and a possible role for other immune cells cannot be ruled out.

Some suggestions from this reviewer are the following:

1. Pathogenic variants of Lrp10 have been associated with Parkinson and ALS. While I understand that the generation and characterization of new point mutation mouse models is of course beyond the scope of this project, it would be interesting to check in a reconstitution experiment (similar to the ones already performed by the authors) the effects of these pathogenic variants on IL7R binding and expression.

Response: We thank the reviewer for pointing us to these studies. Several missense mutations are associated with alpha-synucleinopathies. Dr. Mandemaker's lab has started to test some of these variants functionally and found that Lrp10-R235C was associated with enlarged Lrp10+ vesicles in a microscopy assay looking at brain tissue. We tested the R235C variant and an additional R152C variant (that did not have a microscopy phenotype) in the retroviral reconstitution assay in Lrp10 KO T cells. Each of these variants suppressed IL7R expression to the extent of Lrp10-WT (Fig. EV4B). As an additional control, we reconstituted Lrp10 KO CD8 T cells with Lrp10-D246Y: the variant encoded by the ENU-induced mutation in the original *chowmein* pedigree. Lrp10-D246Y did not suppress IL7R levels. Currently, how D246Y (which is located in the ECD) affects Lrp10's function is not clear. For example, it may affect protein stability or localization. However, these results show that the reconstitution system can assess for the functional effects of coding variants. There are many Lrp10 variants reported that have been associated with alpha-synucleinopathies that we have not yet tested. Reconstitution within the Lrp10 KO system may provide a complementary approach to functional testing.

2. Total body Lrp10 KO in mice is overall protective against tumor because of the CD8 mechanism described by the authors. Are there any publicly available data to generate Kaplan-Meier plot assessing correlation of Lrp10 expression in patients with hot vs cold tumors and their survival?

Response: We have expanded our discussion of possible tumor factors that dictate responsiveness to Lrp10 deletion and cited a study which linked elevated Lrp10 expression to decreased survival in hepatocellular carcinoma, lung adenocarcinoma, and pancreatic adenocarcinoma (PMID 29088295, lines 484-493). Notably, this data is from bulk tumor RNAseq and it is unclear if elevated Lrp10 expression is found in CD8 T cells or in other cell populations. The specific question asked by the reviewer is an important one and establishes the relevance of Lrp10 in humans. However, finding a rigorous and reliable answer requires datasets that ideally have combined scRNAseq data of the tumor infiltrating lymphocytes, assessment of tumor mutational burden, and clinical outcomes. At present, this type of human genomic analysis is beyond the expertise of our lab.

3. The authors are well aware of a possible contribution of other Lrp10 KO cells to the higher tumor infiltrating and clearance potential of Lrp10 CD8 KO cells in the total body KO model. Nevertheless, an experiment that would help solve this question is possible and well in the capabilities of the authors in terms of mouse strains and tumor models available and would consist in adoptively transfer Lrp10^{+/+} and Lrp10^{-/-} CD8 OT-I cells in syngeneic mice previously inoculated with EL4-OVA, B16-OVA and MC38-OVA tumors and check tumor progression or remission.

Response: We thank the reviewer for this suggestion, and we performed these experiments. We adoptively transferred WT or Lrp10 KO OT1 cells into mice inoculated with B16-ova. First, we found that injection of either cell type into tumor-bearing lymphocyte-replete (CD45.1) mice did not impart any protection against tumor progression (Fig. EV5D). This is consistent with most adoptive transfer experiments, which require a lymphocyte depletion step to "clear space" for adoptively transferred lymphocytes. In contrast, when WT or Lrp10 KO OT1 cells were injected into tumor-bearing,

lymphocyte-depleted recipients (Rag2 KO), both types of cells were able to reject established tumors robustly. Therefore, with this experimental approach, we cannot determine if Lrp10 deletion imparts any advantage to primary CD8 T cell anti-tumor response. We noted that Lrp10 KO OT1 cells persisted in the circulation at higher frequencies compared to WT OT1 cells one month after primary challenge (Fig. EV5E). However, mice receiving either WT OT1 or Lrp10 KO OT1 cells were equally resistant to a subsequent tumor challenge, indicating that Lrp10 deletion did not improve anti-tumor memory against B16-ova at one month after initial rejection. It is possible that longer time points after primary rejection may reveal an advantage of the Lrp10 KO recall response. Therefore, the overall question of how much of the tumor resistance phenotype in Lrp10 KO mice is intrinsic to CD8 T cells remains unanswered and will need to be addressed in the future with a conditional knockout strain.

We will note that testing antigen-specific responses against single strong foreign antigens, or immunodominant epitopes, may not accurately reveal the role of Lrp10 in CD8 T cell-mediated anti-tumor immunity. We suspect that a key aspect of Lrp10's function is suppression of clones responding to subdominant antigens thereby curtailing polyclonal responses. We anticipate that studying the role of Lrp10 in CD8 responses using experimental systems that express a diverse antigenic hierarchy in a controlled manner will be informative.

Minor:

4. A reference in the main text to Fig S2B is missing.

Response: This data has become Fig. EV2C with reference in the text.

5. Line 216, S4A refers to % of CD4 cells in the CD45+ subset and this is not precisely mentioned in the text that instead describes S4B and the FoxP3+ in the CD4 subset.

Response: We have combined and re-labeled these two pieces of data in Fig. EV5A to indicate that the FoxP3+ population is within the CD4+ parent population.

Dear Dr. Nair-Gill,

Thank you for the submission of your revised manuscript to our editorial offices. I have now received the reports from the three referees that I asked to re-evaluate the study, you will find below. As you will see, the referees now fully supports the publication of the study in EMBO reports.

Before we can proceed with formal acceptance, I have these final editorial requests:

- Please provide the abstract written in present tense throughout.
- Please add up to 5 keywords to the manuscript and order the manuscript sections like this, using these names: Title page - Abstract - Keywords - Introduction - Results - Discussion - Methods - Data availability section - Acknowledgements - Disclosure and Competing Interests Statement - References - Figure legends - Expanded View Figure legends
- Please remove the file 'Combined figures' (that also seems outdated) from the manuscript files. Just keep the separate files for each final figure (main and EV figures).
- Please make sure that the number "n" for how many independent experiments were performed, their nature (biological versus technical replicates), the bars and error bars (e.g. SEM, SD) and the test used to calculate p-values is indicated in the respective figure legends (also for potential EV figures and all those in the final Appendix). Please also check that all the p-values are explained in the legend, and that these fit to those shown in the figure. Please provide statistical testing where applicable. Please avoid the phrase 'independent experiment', but clearly state if these were biological or technical replicates. Please also indicate (e.g. with n.s.) if testing was performed, but the differences are not significant. In case n=2, please show the data as separate datapoints without error bars and statistics. See also:
<http://www.embopress.org/page/journal/14693178/authorguide#statisticalanalysis>

If $n < 5$, please show single datapoints for diagrams. Presently, most diagrams seem to miss error bars. Please check. Moreover:

- Please indicate the statistical test used for data analysis in the legends of figures 3a; 6a.
- Please note that information related to n is missing in the legends of figures 2e; 6b; EV 2d-f; EV 3d; EV 5e.
- Please note that the error bars are not defined in the legends of figures 4a, d; EV 2f.
- Please add to each legend (main, EV and Appendix figures, where applicable) a 'Data Information' section explaining the statistics used or providing information regarding replicates and scales. See:

- Please move the three tables (EV tables) and their legends to the Appendix. Please name these Appendix Tables S1-3 and change their callouts accordingly. Finally, please remove the EV table legends from the manuscript text file.
- Please correct the name of the Appendix figure to Appendix Figure S1.
- Please make sure that all the funding information is also entered into the online submission system and that it is complete and similar to the one in the acknowledgement section of the manuscript text file. The research agreement with ImmunoDesigners, Inc., is presently only mentioned in the acknowledgements section. Please check.
- Thanks for providing the source data (SD). However, it seems all the numerical data for the diagrams shown in the main figures and some other panels is missing (i.e. for Fig. 1 B,E,F,G,I,J,K,L; Fig. 2 A,C,D,E; Fig. 3A,B; Fig. 4A-G; Fig. 5A,C,F,G,H; and Fig. 6A-C,E-I). Please carefully go again through the source data checklist and make sure all the requested SD is provided. Moreover, please upload the final source data as one ZIPed folder per figure.

In addition, I would need from you:

Best,

Referee #1:

The authors have addressed my concerns.

Referee #2:

I am satisfied with the answers and the revised manuscript from the authors.

Referee #3:

No further comments. The authors addressed this reviewer's questions.

All editorial and formatting issues were resolved by the authors.

Dr. Evan Nair-Gill
UT Southwestern
Center for the Genetics of Host Defense
5323 Harry Hines Blvd
Dallas, TX 75390
United States

Dear Dr. Nair-Gill,

I am very pleased to accept your manuscript for publication in the next available issue of EMBO reports. Thank you for your contribution to our journal.

Yours sincerely,
